# A versatile, fast and unbiased method for estimation of gene-by-environment interaction effects on biobank-scale datasets

Matteo Di Scipio[1,2,10], Mohammad Khan [1,2,10], Shihong Mao[1], Michael Chong[1,3,4], Conor Judge[1], Nazia Pathan [1,2], Nicolas Perrot [1], Walter Nelson [5,6], Ricky Lali[1,7], Shuang Di[5,8], Robert Morton[1,4], Jeremy Petch [1,2,5,9] & Guillaume Paré [1,3,4,7] ✉

Identification of gene-by-environment interactions (GxE) is crucial to understand the interplay of environmental effects on complex traits. However, current methods evaluating GxE on biobank-scale datasets have limitations. We introduce MonsterLM, a multiple linear regression method that does not rely on model specification and provides unbiased estimates of variance explained by GxE. We demonstrate robustness of MonsterLM through comprehensive genome-wide simulations using real genetic data from 325,989 individuals. We estimate GxE using waist-to-hip-ratio, smoking, and exercise as the environmental variables on 13 outcomes (N = 297,529-325,989) in the UK Biobank. GxE variance is significant for 8 environment-outcome pairs, ranging from 0.009 – 0.071. The majority of GxE variance involves SNPs without strong marginal or interaction associations. We observe modest improvements in polygenic score prediction when incorporating GxE. Our results imply a significant contribution of GxE to complex trait variance and we show MonsterLM to be well-purposed to handle this with biobank-scale data.

Identifying gene-by-environment interactions (GxE) is difficult because individual interaction effects are expected to be small[1], the multiple hypothesis burden is considerable[2,3], and the sample sizes needed are correspondingly large ($N > 300,000$)[4]. Many previous analyses have focused on identifying interactions with variants marginally associated with a phenotype of interest[5,6]. Hitherto, methods developed to estimate the overall effect of these interactions rely on variance component methods, due to the predictor ($m$) > observation ($n$) problem, where SNPs ($m$) vastly outnumber the participants ($n$)[7,8]. These methods are advantageous for smaller datasets; however, they can be limiting when applied to larger datasets due to computational burden[7]. Furthermore, variance component methods depend on strong assumptions about the underlying genetic model and often require a priori specification of parameters and/or hyper-parameters, such as polygenicity, minor allele frequency (MAF), and linkage disequilibrium (LD) dependence[9–13]. While never formally tested in the context of GxE, it has previously been shown that these assumptions can lead to

[1]Population Health Research Institute, David Braley Cardiac, Vascular and Stroke Research Institute, Hamilton Health Sciences and McMaster University, Hamilton, ON, Canada. [2]Department of Medicine, Faculty of Health Sciences, McMaster University, Hamilton, ON, Canada. [3]Thrombosis and Atherosclerosis Research Institute, David Braley Cardiac, Vascular and Stroke Research Institute, Hamilton, ON, Canada. [4]Department of Pathology and Molecular Medicine, McMaster University, Michael G. DeGroote School of Medicine, Hamilton, ON, Canada. [5]Centre for Data Science and Digital Health, Hamilton Health Sciences, Hamilton, ON, Canada. [6]Department of Statistical Sciences, University of Toronto, Toronto, ON, Canada. [7]Department of Health Research Methods, Evidence, and Impact, McMaster University, Hamilton, ON, Canada. [8]Dalla Lana School of Public Health, University of Toronto, Toronto, ON, Canada. [9]Institute of Health Policy, Management and Evaluation, University of Toronto, Toronto, ON, Canada. [10]These authors contributed equally: Matteo Di Scipio, Mohammad Khan. ✉e-mail: pareg@mcmaster.ca

important biases in heritability estimates[9–11,14–16]. Novel methods are thus needed to enable fast and unbiased calculations of the variance explained ($R^2$) by GxE in large samples, on multiple traits and without the need for genetic model assumptions.

Our proposed method is similar to the generalized random effects (GRE) model[17], building on the observation that the multiple regression coefficient of determination can be used to accurately estimate heritability[17]. Extending this observation to include an environmental exposure variable and computing the interactions between genotypes and the environmental exposure allows us to examine the variance explained by genetic interactions with an environmental exposure. However, the large number of single nucleotide polymorphisms (SNPs) ($m$) compared to participants ($n$) presents a challenge for genome-wide analysis[18]. By partitioning the genome into non-overlapping regions, it becomes possible to estimate genome-wide interactions with environmental exposures by reducing $m$ within each region to a size where $m < n$. Some challenges remain: First, LD spillage at the junction of blocks can theoretically inflate heritability estimates if many such junctions exist[9]. Second, any residual population stratification effects would be

amplified if heritability at each region is overestimated and this effect is expected to be proportional to the number of blocks[19]. Third, computing prediction $R^2$ on large blocks with high dimensionality can be slow. By using the conjugate gradient method[20] with graphics processing unit (GPU) acceleration[21], it is possible to perform multiple linear regression modelling efficiently on large (25,000 SNPs) blocks. Thus, the potential for residual population stratification effects and LD spills are minimized since only 60 blocks or less are needed for genome-wide analyses and the variants included are LD-pruned. Furthermore, a block size of no more than 25,000 SNPs also ensures that $n > 10\,m$ for accurate estimations.

In this work we propose MonsterLM, a method to estimate the proportion of variance explained by GxE, in a fast, accurate, efficient, and unbiased manner on biobank-scale datasets ($N > 300,000$). We hypothesize that GxE interactions contribute significantly to complex trait variance. Our objective is to quantify and characterize these contributions for 13 complex traits using four environmental exposures (waist-to-hip ratio [WHR], smoking, an exercise parameter, and a randomly generated exposure). We illustrate an overview of our computational analyses in Fig. 1.

## GxE Interaction Variance Estimation (MonsterLM)

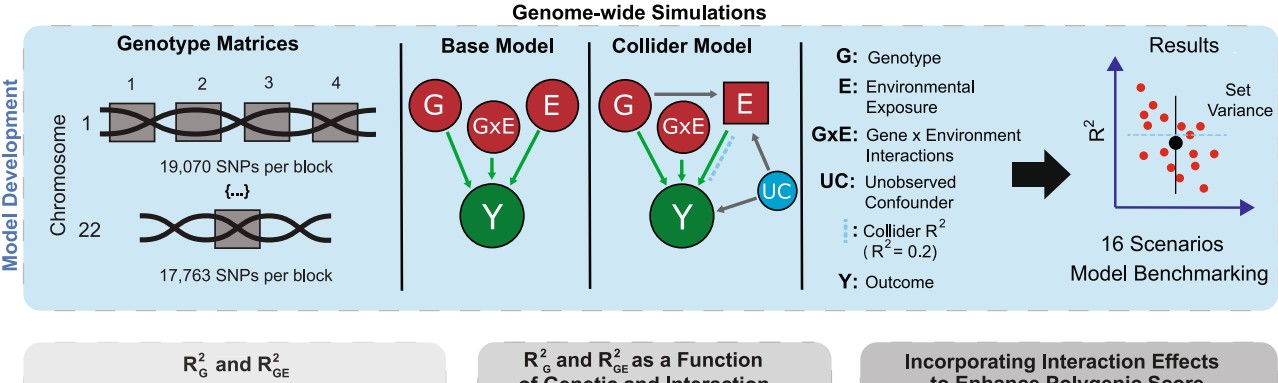

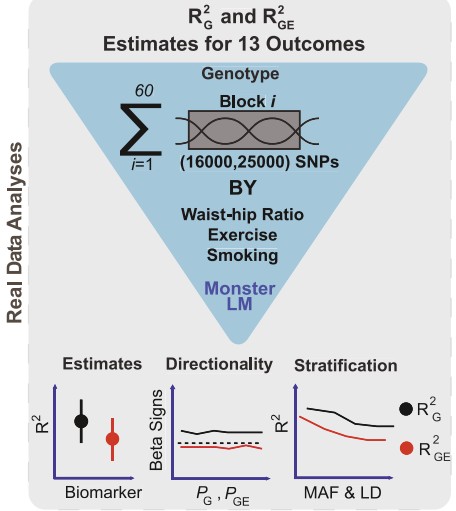

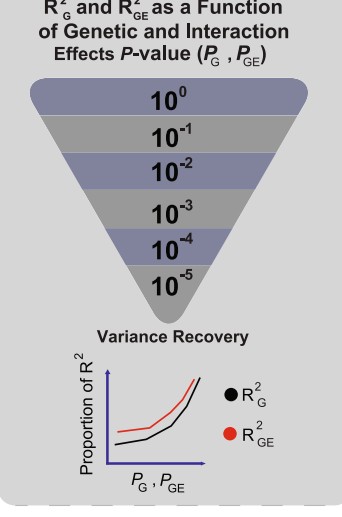

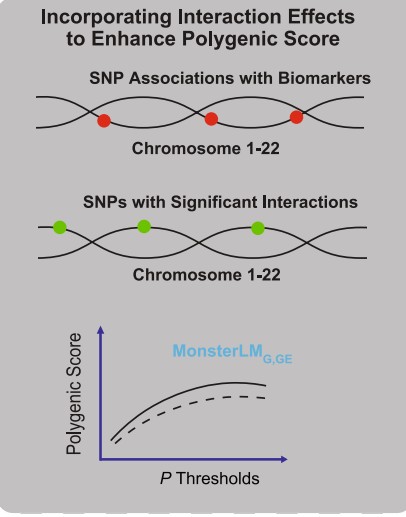

**Fig. 1 | Summary of gene-by-environment (GxE) analysis conducted with MonsterLM.** Initial simulation studies were conducted to verify the properties of MonsterLM. Simulated outcomes with known values for variance explained were regressed under varying scenarios and model specifications to ensure robust estimation (blue panel). Real outcome analyses were conducted with UK Biobank data (grey panels). Genome-wide SNP heritability estimates with and without waist-hip-ratio (WHR) interactions revealed significant interaction effects for 8 of 13 outcomes and were further assessed with a directionality of effects and stratification analysis (bottom left panel). MonsterLM properties were further explored recovering genotype and interaction variance explained through partitioning SNPs based on genotype and interaction univariate regressions (bottom middle panel). Lastly, sequential incorporation of subsets of SNPs with significant interaction associations derived from univariate interaction regressions of the genotype SNPs on their respective outcomes revealed modest improvements of polygenic scores in one of the eight outcomes tested (bottom right panel). Genotype Matrices: Block partitioning schematic. Base and Collider Models: green circles are outcome variables; red circles are predictor variables; red squares are colliders; blue circles are confounders; green arrows are causal associations; grey arrows are unobserved causal associations. $R^2_G$: Genetic variance; $R^2_{GE}$: GxE variance; $P_G$: univariate genetic association SNPs; $P_{GE}$: univariate interaction association SNPs.

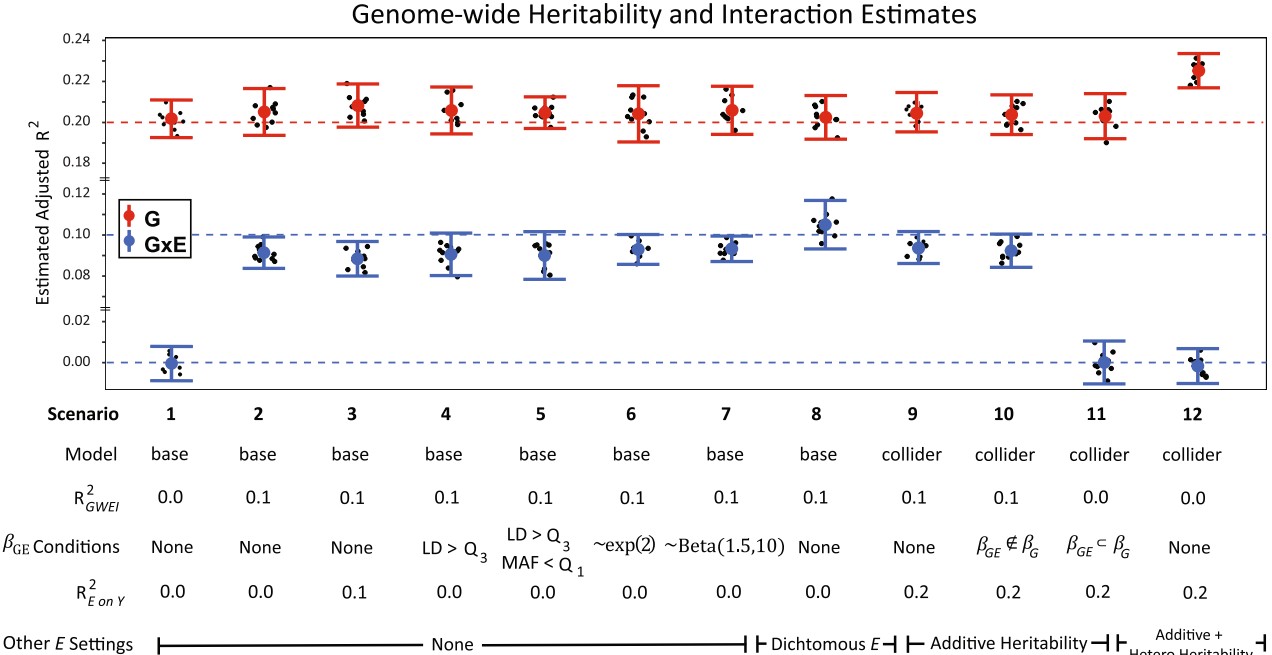

**Fig. 2 | Estimation of variance explained by GxE for 12 simulated scenarios.**
Estimation of variance explained by GxE for 12 genome-wide scenarios from 10 simulations. "None" indicates the absence of a condition. *Model*: with or without collider features. Dashed red lines indicate true set G ($R^2_{GW}$) variance and dashed blue lines indicate true set GxE variance ($R^2_{GWEI}$). $\beta_{GE}$ *Conditions: LD > Q_3*: all $\beta_{GE}$ effects were sampled from GxE effect SNPs in the highest LD quartile; *MAF < Q_1*: all $\beta_{GE}$ effects were sampled from GxE effect SNPs in the lowest MAF quartile; sampling distribution for $\beta_{GE}$ other than -$N(0,1)$ is denoted; $R^2_{E\ on\ Y}$: outcome variance explained by exposure; *E* continuous unless otherwise stated; *E* Heritability: additive or heterogeneous. Scenario conditions toggle these parameters: (i) estimation in the null base scenario ($R^2_{GWEI} = 0$), (ii) estimation in the non-null base scenario ($R^2_{GWEI} = 0.1$), (iii) estimation when the exposure variance is raised to 0.1, (iv) estimation when $\beta_{GE}$ is sampled from *LD SNPs > Q_3*, (v) estimation when $\beta_{GE}$ is sampled from *LD* SNPs > $Q_3$ and *MAF* SNPs < $Q_1$, (vi–vii) estimation when the

assumptions of standardization for $\beta_{GE}$ effects were invalidated by generating effects with exponential and beta distributions (positive kurtosis), (viii) estimation in scenario (i) but using a dichotomous generated $E_{sim}$, (ix) estimation in the collider scenario where $\beta_{GE}$ and $\beta_G$ effects were randomly selected, (x) estimation in the collider scenario where $\beta_{GE}$ effects were not an element (completely non-overlapping) of $\beta_G$ effects, (xi) estimation in the collider scenario where $\beta_{GE}$ effects are a strict subset (completely overlapping) of $\beta_G$ effects, and (xii) estimation in the collider scenario where simulated exposures are heritable through additive and heterogenous genetic effects. Means and 95% confidence intervals are represented by dot and whisker plots per scenario. Each black dot represents a single genome-wide simulation. Simulations were based on quality controlled UKB data consisting of 325,989 individuals and 1,030,579 SNPs. Source data are provided as a Source Data file.

## Results
### Validation of MonsterLM using simulations
We conducted ten genome-wide simulations for each of the 12 scenarios (Fig. 2). The true heritability ($R^2_{GW}$) was set to 0.20 and the true interaction variance ($R^2_{GWEI}$) was set to 0.1 or 0.0. MonsterLM accurately and precisely estimated the true $R^2_{GWEI}$ across all 12 scenarios (Fig. 2). Under the null scenarios of no GxE, the estimated $R^2_{GWEI}$ was not different from zero ($p > 0.05$) in all ten simulations. Furthermore, observed precision estimates (i.e. variance of estimates across the 12 simulations) did not significantly differ from the precision estimates predicted from Eqs. (8)–(11) (Supplementary Table 1).

MonsterLM accurately detected G and GxE null and non-null effects when $R^2_{GWEI}$ was set to 0.0 or 0.1, with $R^2_{GW}$ fixed at 0.20 (Fig. 2; scenario 1–2). These results remained when a true causal effect of $E_{sim}$ on $Y_{sim}$ ($R^2_{E\ on\ Y} = 0.1$) was simulated (Fig. 2; scenario 3). MonsterLM also remained unbiased to varying distributions of GxE effects, where non-zero GxE effects (i.e. $\beta_{GE} \neq 0$) were sampled from exponential and beta (positive kurtosis) distributions (Fig. 2; scenario 6–7). Accurate GxE estimations were observed in the four scenarios including collider biases (Fig. 2; scenario 9–12). Accurate G estimations were observed in all collider scenarios except when exposure heritability was heterogeneous (Fig. 2; scenario 9–12).

To assess the robustness of MonsterLM to LD, SNPs with non-zero GxE effects were exclusively selected from SNPs in the highest quartile of LDscore or from SNPs in the highest quartile of LDscore and lowest quartile of MAF (Fig. 2; scenario 4–5). No significant bias in G or GxE

was observed. We further stratified SNPs into 20 bins based on MAF and LD, and individually tested each stratum for G and GxE effects. Each stratum provided consistent estimates (after adjusting for the number of SNPs), further confirming the robustness of MonsterLM to MAF and LD (Supplementary Table 2).

Simulation estimates remained unbiased with dichotomous exposures (Fig. 2; scenario 7) and dichotomous outcomes (Supplementary Table 3) when applying MonsterLM with the modifications for dichotomous variables outlined in the methods.

Lastly, the performance of MonsterLM was tested in simulations where missing exposures or outcomes were mean imputed. Using the settings of scenario 2, the GxE estimate was biased towards the null when 20% of the exposure *or* 20% of the outcome in randomly chosen individuals were mean imputed (Supplementary Table 4). However, if 20% of the exposure *and* 20% of the outcome were missing in the same individuals and subsequently mean imputed, then the GxE estimates were inflated (Supplementary Table 4). Hence, all analyses using real data were performed on participants with no missing data.

MonsterLM power was evaluated with simulations for $R^2_{GWEI}$ varying from 0.005 to 0.50 and sample sizes ranging from 50,000 to 400,000 participants (Supplementary Fig. 1). MonsterLM reliably detects G and GxE effects with a minimum of 80% power when $N > 100,000$ participants and the true $R^2 > 0.05$. At a biobank sample size of 325,000, MonsterLM is well powered to detect true $R^2 > 0.01$.

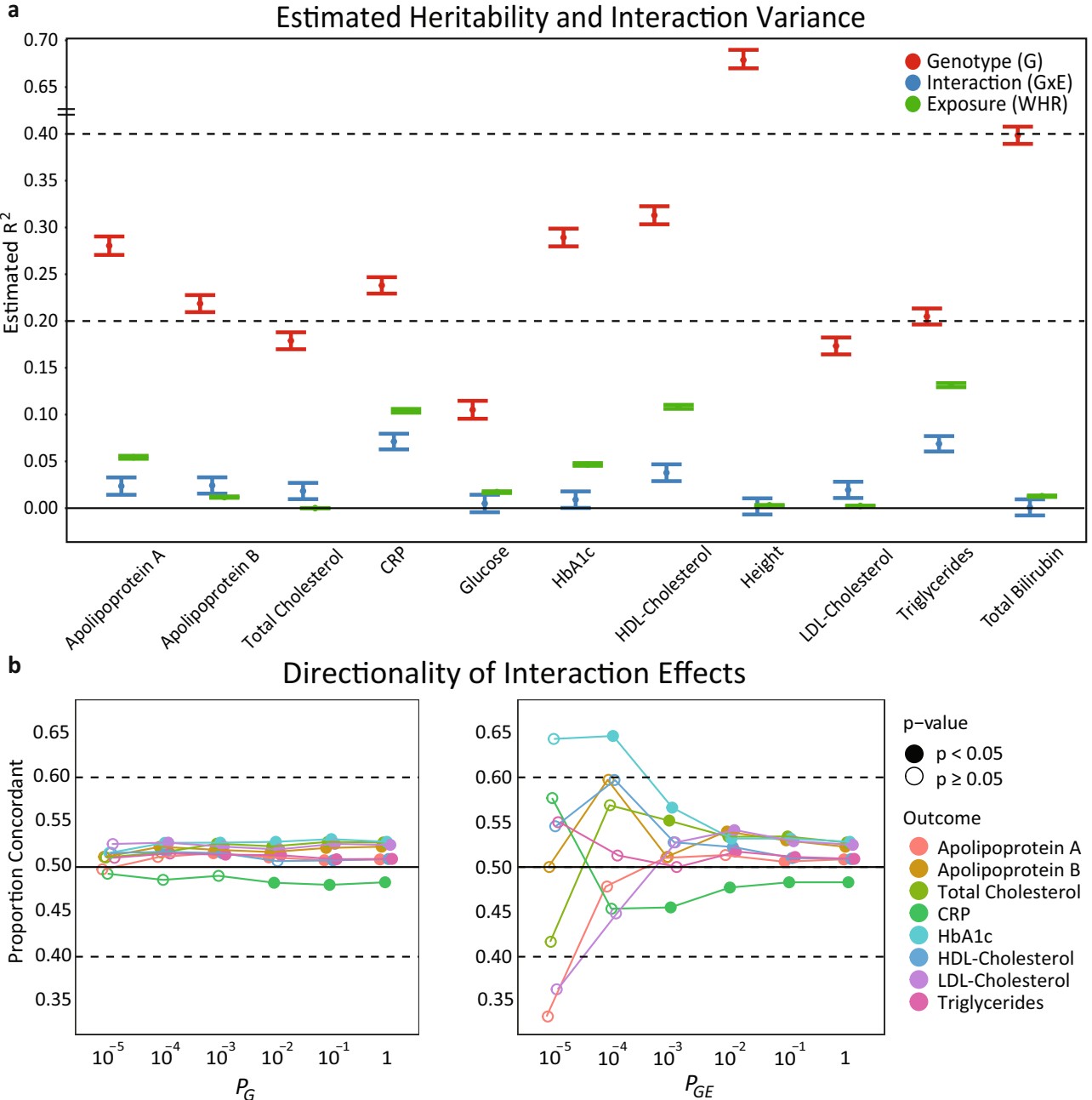

**Fig. 3 | Estimates of genetic, interaction, and environment (WHR) $R^2$.** Estimates were computed for eleven outcomes with associated directionality of effects using the MonsterLM methodology. **a** Genetic, interaction, and environment (WHR) variance estimated $R^2$ for each outcome using the MonsterLM protocol. Estimates and 95% confidence intervals are represented by dot and whisker plots. **b** The directionality of effects for derived interaction estimates. SNPs were filtered based on univariate $P_G$, $P_{GE}$ and LD ($r^2 < 0.1$) for each outcome. Directionality is concordant when $\hat{\beta}_G$ and $\hat{\beta}_{GE}$ have the same sign (+/+, −/−) and discordant when they have opposite signs (+/−, −/+). Two-proportion $Z$-tests were used to compare each directionality result with a null value of 0.5. Two-sided significance was defined as $p < 0.05$. Directionality was computed only for significant outcomes. Estimates were conducted with 325,989 individuals and 1,030,579 SNPs after quality control. Source data are provided as a Source Data file.

## Genome-wide interaction and heritability estimation in the UK Biobank

We applied MonsterLM to estimate the GxE variance between three environmental exposures (WHR, days of at least 10 minutes moderate physical activity status [M10], and smoking status) and ten cardiometabolic blood biomarkers (Apolipoprotein A, Apolipoprotein B, Total Cholesterol, CRP, Glucose, HbA1c, HDL-Cholesterol, LDL-Cholesterol, Triglycerides, Total Bilirubin), two cardiometabolic diseases (Coronary Artery Disease [CAD] and Type 2 Diabetes [T2D]), and height. Of the 13 outcomes, we observed significant GxE with WHR for 8 outcomes ($R^2_{GWEI}$ ranging from 0.009 to 0.071), significant GxE with M10 for 7 outcomes ($R^2_{GWEI}$ ranging from 0.010 to 0.045), and no significant GxE with smoking for any outcomes (Fig. 3a; Tables 1 and 2). The strongest GxE with WHR was observed for CRP ($R^2_{GWEI} = 0.071$) and strongest GxE

**Table 1 | Real data estimates for continuous outcomes with MonsterLM**

| Trait | Additive genetic variance (95% CI) | WHR GxE variance (95% CI) | M10 GxE variance (95% CI) | Smoking GxE variance (95% CI) | Permuted exposure GxE (95% CI) |
|---|---|---|---|---|---|
| Apolipoprotein A | **0.281** (0.271, 0.290) | **0.0236** (0.014, 0.033) | 0.007 (−0.003, 0.017) | 0.000739 (−0.0193, 0.0207) | −0.0027 (−0.0127, 0.0073) |
| Apolipoprotein B | **0.219** (0.210, 0.228) | **0.0242** (0.016, 0.033) | **0.040** (0.030, 0.050) | 0.00984 (−0.0102, 0.0298) | −0.0038 (−0.0134, 0.0062) |
| Total Cholesterol | **0.179** (0.170, 0.188) | **0.0183** (0.010, 0.027) | **0.042** (0.032, 0.052) | 0.00505 (−0.0150, 0.0251) | 0.0000 (−0.0099, 0.0099) |
| CRP | **0.238** (0.229, 0.247) | **0.0711** (0.063, 0.080) | **0.028** (0.018, 0.038) | 0.0100 (−0.010, 0.030) | −0.0022 (−0.0122, 0.0078) |
| Glucose | **0.105** (0.096, 0.115) | 0.00475 (−0.004, 0.014) | 0.003 (−0.007, 0.013) | 0.00776 (−0.0122, 0.0278) | −0.0007 (−0.0106, 0.0092) |
| HDL-Cholesterol | **0.313** (0.303, 0.323) | **0.0380** (0.029, 0.047) | **0.010** (0.001, 0.020) | −0.00234 (−0.0223, 0.0177) | −0.0020 (−0.0120, 0.0080) |
| HbA1c | **0.289** (0.280, 0.299) | **0.00884** (0.000, 0.018) | **0.013** (0.003, 0.023) | 0.00610 (−0.0139, 0.0261) | −0.0010 (−0.0110, 0.0006) |
| Height | **0.683** (0.674, 0.694) | 0.00136 (−0.007, 0.010) | 0.0047 (−0.005, 0.015) | −0.00124 (−0.0212, 0.0188) | −0.0013 (−0.0089, 0.0063) |
| LDL-Cholesterol | **0.173** (0.164, 0.182) | **0.0194** (0.011, 0.028) | **0.045** (0.035, 0.055) | 0.00825 (−0.0118, 0.0283) | 0.0014 (−0.0085, 0.0114) |
| Triglycerides | **0.205** (0.196, 0.213) | **0.0680** (0.061, 0.077) | **0.024** (0.014, 0.034) | 0.00696 (−0.0130, 0.0270) | −0.0006 (−0.0106, 0.0094) |
| Total Bilirubin | **0.399** (0.389, 0.408) | 0.000634 (−0.008, 0.009) | 0.002 (−0.007, 0.012) | 0.00498 (−0.0150, 0.0250) | −0.0019 (−0.0119, 0.0081) |

MonsterLM real data results for continuous outcomes. Presented are real data estimates for continuous outcomes. All estimates are performed using the MonsterLM methodology. Exposures include waist-hip-ratio (WHR), number of days of 10 minutes moderate exercise (M10), dichotomous smoking status (Smoking), and permuted exposure ($E_{pm}$). Bolded estimates are significant.

**Table 2 | Real data estimates for dichotomous outcomes with MonsterLM**

| Trait | Additive genetic variance (95% CI) | WHR GxE variance (95% CI) | M10 GxE variance (95% CI) | Permuted exposure GxE (95% CI) |
|---|---|---|---|---|
| Type 2 diabetes | **0.659** (0.562, 0.755) | −0.0015 (−0.0385, 0.0355) | 0.0173 (−0.00265, 0.0378) | −0.0201 (−0.0721, 0.0319) |
| Coronary artery disease | **0.181** (0.144, 0.218) | −0.0057 (−0.0407, 0.0293) | −0.0105 (−0.0405, 0.0101) | −0.0331 (−0.0785, 0.0123) |

MonsterLM real data results for dichotomous outcomes. Presented are real data estimates for dichotomous outcomes. All estimates are performed using the MonsterLM methodology. Exposures include waist-hip-ratio (WHR), number of days of 10 minutes moderate exercise (M10), and a randomly permuted exposure ($E_{pm}$). Bolded estimates are significant.

with M10 was observed for LDL-Cholesterol ($R^2_{GWEI} = 0.045$). For some outcomes, interactions explained a substantial fraction of variance relative to heritability. For example, GxE with WHR explained 33% and 27% as much variance in Triglycerides and CRP as its estimated heritability, respectively. Generally, GxE with M10 results displayed consistent albeit attenuated $R^2_{GWEI}$ compared with GxE with WHR (Table 1). No significant GxE variance was observed for dichotomous outcomes CAD and T2D (Table 2) nor for randomly permuted exposures for all outcomes (Tables 1 and 2).

Outcome heritability estimates for all 13 traits were significant and largely consistent with published estimates and other methods (BOLT, mtg2, and GRE; Tables 2 and 3). For two dichotomous outcomes, CAD and T2D (Table 2), genetic variance was estimated at 0.181 and 0.659, respectively, on the liability scale, which is consistent with the reported heritability of these diseases in literature[22–25]. As MonsterLM adjusts outcomes for each specific exposure tested and this could potentially impact heritability estimates (which do not necessarily require such exposure adjustments) the analysis was repeated without adjustment, with consistent results (Supplementary Table 5).

Follow-up analyses for GxE with WHR were performed to further observe components of the method. We observed significant directionality for interaction effects at both univariate marginal and interaction association $p$-values, $P_G$ and $P_{GE}$, for multiple $p$-value thresholds ($<10^{-3}$, $<10^{-2}$, $<10^{-1}$, and $<1$; Fig. 3b; Supplementary Figure 2). That is, signs of $\hat{\beta}_G$ and $\hat{\beta}_{GE}$ were the same (+/+ or −/−) more often than expected in the null condition (>50%) for most outcomes when sorted by $P_G$ and $P_{GE}$. Consistent with the directionality concordance for each outcome at $P_G < 1$ and $P_{GE} < 1$, Pearson correlation coefficients of estimated genetic regression coefficients for each outcome, $\hat{\beta}_{1 \times m}$ (m is the number of SNPs: 1,030,579), were significant for all outcomes in Fig. 3b for $\hat{\beta}_G$ and $\hat{\beta}_{GE}$ (Supplementary Table 6). When extending the Pearson correlation tests to estimated genetic regression coefficients from WHR heritability (WHR heritability; Supplementary Table 5), $\hat{\beta}_{h^2_{WHR}}$, neither $\hat{\beta}_G$ or $\hat{\beta}_{GE}$ were significantly correlated with $\hat{\beta}_{h^2_{WHR}}$ for

almost all outcomes (Supplementary Table 6). To assess the uniformity of the contribution of SNPs to both G and GxE, we stratified SNPs into twenty categories based on MAF and LDscore (Supplementary Table 2). The average SNP contribution to G and GxE did not markedly differ by MAF or LDscore, confirming the absence of large differences in contribution (Supplementary Fig. 3).

**Comparison with other methods**
Heritability estimates were largely consistent between MonsterLM, BOLT, mtg2, and GRE (Table 3). One notable exception was for total bilirubin, for which heritability was overestimated by GRE ($R^2_{GW} > 0.99$) relative to MonsterLM ($R^2_{GW} = 0.40$), BOLT ($R^2_{GW} = 0.37$) and mtg2 ($R^2_{GW} = 0.43$). mtg2 and LDSC heritability estimates were lower compared to MonsterLM, Bolt, and GRE for all compared outcomes except total bilirubin (with mtg2).

MonsterLM GxE estimates with WHR were compared to mtg2 and LDSC (Table 3). The mtg2 analysis was limited to 75,000 individuals due to computational constraints. GxE estimates were consistent between MonsterLM and mtg2 for cholesterol, height, and total bilirubin. However, glucose and HbA1c had considerably higher GxE estimates with mtg2 compared to MonsterLM (0.210 and 0.139 in mtg2, versus 0.00475 and 0.00884 in MonsterLM). MonsterLM heritability estimates of glucose and HbA1c were more consistent with Bolt and GRE versus mtg2 (0.105 and 0.289 in MonsterLM, versus 0.045 and 0.129 in mtg2). The LDSC GxE analysis used the full participant list, SNP set, and phenotypic data as in MonsterLM. GxE estimates were lower than MonsterLM for all outcomes.

The comparison between MonsterLM and mtg2 was further extended to simulated outcomes and exposures, with ten simulations split between true set $R^2_{GWEI} = 0.10$ and $R^2_{GWEI} = 0$ (Supplementary Table 7). MonsterLM accurately estimated the true interaction variance explained in all ten simulations. mtg2 was accurate in most simulations but overestimated the set interaction variance in three

**Table 3 | Benchmarking MonsterLM additive genetic variance (heritability, $h^2_G$) and GxE (GxE, $R^2_{GxE_{WHR}}$)**

| Trait | MonsterLM $h^2_G$ ($\sigma$) | mtg2 $h^2_G$ ($\sigma$) | LDSC $h^2_G$ ($\sigma$) | Bolt $h^2_G$ ($\sigma$) | GRE $h^2_G$ ($\sigma$) | MonsterLM $R^2_{GxE_{WHR}}$ ($\sigma$) | mtg2 $R^2_{GxE_{WHR}}$ ($\sigma$) | LDSC $R^2_{GxE_{WHR}}$ ($\sigma$) |
|---|---|---|---|---|---|---|---|---|
| Apolipoprotein A | 0.281 (0.0046) | 0.220 (0.0061) | 0.217 (0.0320) | 0.240 (0.0025) | 0.295 (0.005) | 0.024 (0.0043) | 0.005 (0.0027) | 0.003 (0.0021) |
| Apolipoprotein B | 0.219 (0.0045) | 0.194 (0.0069) | 0.129 (0.0246) | 0.232 (0.0023) | 0.274 (0.005) | 0.024 (0.0043) | 0.004 (0.0030) | 0.010 (0.0020) |
| Cholesterol | 0.179 (0.0045) | 0.157 (0.0065) | 0.122 (0.0181) | 0.160 (0.0022) | 0.198 (0.005) | 0.018 (0.0042) | 0.010 (0.0032) | 0.014 (0.0023) |
| CRP | 0.238 (0.0046) | 0.0706 (0.0060) | 0.056 (0.0079) | 0.228 (0.0023) | 0.283 (0.005) | 0.071 (0.0044) | 0.001 (0.0031) | 0.003 (0.0020) |
| Glucose | 0.105 (0.0047) | 0.0447 (0.0053) | 0.045 (0.0044) | 0.0903 (0.0019) | 0.116 (0.0049) | 0.005 (0.0046) | 0.210 (0.0052) | 0.009 (0.0041) |
| HbA1c | 0.289 (0.0047) | 0.129 (0.006) | 0.113 (0.0072) | 0.246 (0.0024) | 0.284 (0.0051) | 0.009 (0.0044) | 0.139 (0.0045) | 0.018 (0.0041) |
| HDL-Cholesterol | 0.313 (0.0044) | 0.232 (0.0057) | 0.246 (0.0329) | 0.279 (0.0024) | 0.326 (0.0052) | 0.038 (0.0041) | 0.006 (0.0024) | 0.007 (0.0021) |
| LDL-Cholesterol | 0.173 (0.0045) | 0.153 (0.0066) | 0.115 (0.0226) | 0.157 (0.0022) | 0.195 (0.0050) | 0.019 (0.0043) | 0.007 (0.0031) | 0.014 (0.0023) |
| Triglycerides | 0.205 (0.0042) | 0.187 (0.0060) | 0.163 (0.0189) | 0.229 (0.0023) | 0.264 (0.0051) | 0.068 (0.0041) | 0.021 (0.0029) | 0.018 (0.0041) |
| Height | 0.683 (0.0045) | 0.316 (0.0062) | 0.447 (0.0202) | 0.548 (0.0022) | 0.497 (0.0054) | 0.001 (0.0039) | −0.019 (0.0020) | 0.001 (0.0020) |
| Total bilirubin | 0.399 (0.0046) | 0.431 (0.0077) | 0.087 (0.0348) | 0.369 (0.0023) | 1.346 (NA) | 0.001 (0.0043) | 0.000 (0.0028) | 0.000 (0.0017) |

MonsterLM Benchmarking. MonsterLM is compared to other heritability models. Presented are both additive genetic variance and GxE_WHR estimates with calculated standard deviations for the outcomes used in this study. The same genotypic and phenotypic individual-level data ($N = 325{,}989$) is used except for mtg2 which was calculated on a smaller subset of $N = 75{,}000$.

instances (i.e. estimate of 0.125 versus true interaction variance of 0.1 or 0.0).

## Recovery of interaction and heritability variance according to SNP marginal and interaction effects

The presence of significant GxE with WHR prompted additional questions. First, do GxE interactions arise from SNPs strongly associated with the outcome of interest, as has been commonly assumed, or are the variants contributing to GxE interactions independent from those with marginal effects? To address this question, we randomly split participants into a discovery set comprising 80% of participants (260,768 individuals) with the remaining 20% of participants (65,222 individuals) comprising the validation set. Using the eight outcomes with significant GxE interaction variance (no further analyses were conducted with outcomes having non-significant GxE interaction variance), we conducted univariate linear regression on the discovery set using each outcome and a single SNP as the predictor variable, repeating this process for all SNPs (i.e. 1,030,579 SNPs) used in the study and calculating $P_G$, the association $p$-value. We then selected SNPs according to six association $P_G$ thresholds: <1 (i.e. all SNPs), $<10^{-1}$, $<10^{-2}$, $<10^{-3}$, $<10^{-4}$, $<10^{-5}$. Likewise, we tested each SNP individually for interaction with WHR in the discovery set. We selected interactions based on six discovery $P_{GE}$ thresholds: <1 (i.e. all SNPs), $<10^{-1}$, $<10^{-2}$, $<10^{-3}$, $<10^{-4}$, $<10^{-5}$. Each SNP set was then tested for variance explained with the corresponding outcome in the validation set, using the least number of blocks possible while keeping $n > 10\,m$. We evaluated $R^2_{GW}$ and $R^2_{GWEI}$ for each of the eight significant outcomes at each of the six association $P_G$ or $P_{GE}$ thresholds in the SNP validation sets. $R^2_{GW}$ and $R^2_{GWEI}$ were then compared to the variance explained when including all SNPs (i.e. $P_G < 1$ or $P_{GE} < 1$) in the validation set. We estimated the proportion of $R^2_{GW}$ or $R^2_{GWEI}$ in the validation set ($R^2_{G-val.}$ and $R^2_{GE-val.}$) recovered when including an increasing proportion of SNPs in the analysis (Fig. 4; Supplementary Figs. 4 and 5).

We observed that between 42–67% of the original $R^2_{G-val.}$ calculated in the validation set could be recovered with strongly associated marginal SNPs, defined as $P_G < 10^{-5}$ in the discovery set. When extending to more weakly associated SNPs ($P_G < 10^{-1}$), we observed that between 72–89% of $R^2_{G-val.}$ was recovered. Additionally, $R^2_{G-val.}$ recovery when calculating from SNPs with strong or weak interactions in the discovery sample ($P_{GE} < 10^{-5}$ and $P_{GE} < 10^{-1}$, respectively) was consistently lower as compared to their respective $P_G$ thresholds (Fig. 4a).

We then similarly estimated the proportion of interaction variance ($R^2_{GWEI}$) recovered when including an increasing proportion of SNPs, based on discovery $P_G$ or $P_{GE}$ (Fig. 4b; Supplementary Figs. 4 and 5). We observed that between 1–2% of the original $R^2_{GE-val.}$ calculated in the validation set was recovered by SNPs with strong interactions in the discovery set ($P_{GE} < 10^{-5}$). Conversely, 3–28% of the original $R^2_{GE-val.}$ was recovered by SNPs with strong marginal associations ($P_G < 10^{-5}$). When extending to more weakly associated SNPs, between 15–84% of $R^2_{GE-val.}$ was recovered by SNPs with weak marginal associations ($P_G < 10^{-1}$); and between 30–84% of $R^2_{GE-val.}$ was recovered by SNPs with weak interactions ($P_{GE} < 10^{-1}$).

## Polygenic scores analysis

Finally, we examined if the predictiveness of polygenic score of all eight outcomes with significant WHR interaction variance could be improved by incorporating interactions. To select SNPs and interaction effects to be included in each PS, we used both $P_G$ and $P_{GE}$ thresholds of $10^{-2}$, $10^{-3}$, $10^{-4}$, and $10^{-5}$ in the discovery set when testing either each SNP individually or both a single SNP and corresponding interaction, respectively. Each PS was then tested in the validation sample for association with its corresponding outcome, with twenty total $P_G$ and $P_{GE}$ combinations. PS prediction $R^2$ was slightly improved ($p < 0.05$ for improvement) by incorporation of interaction effects for

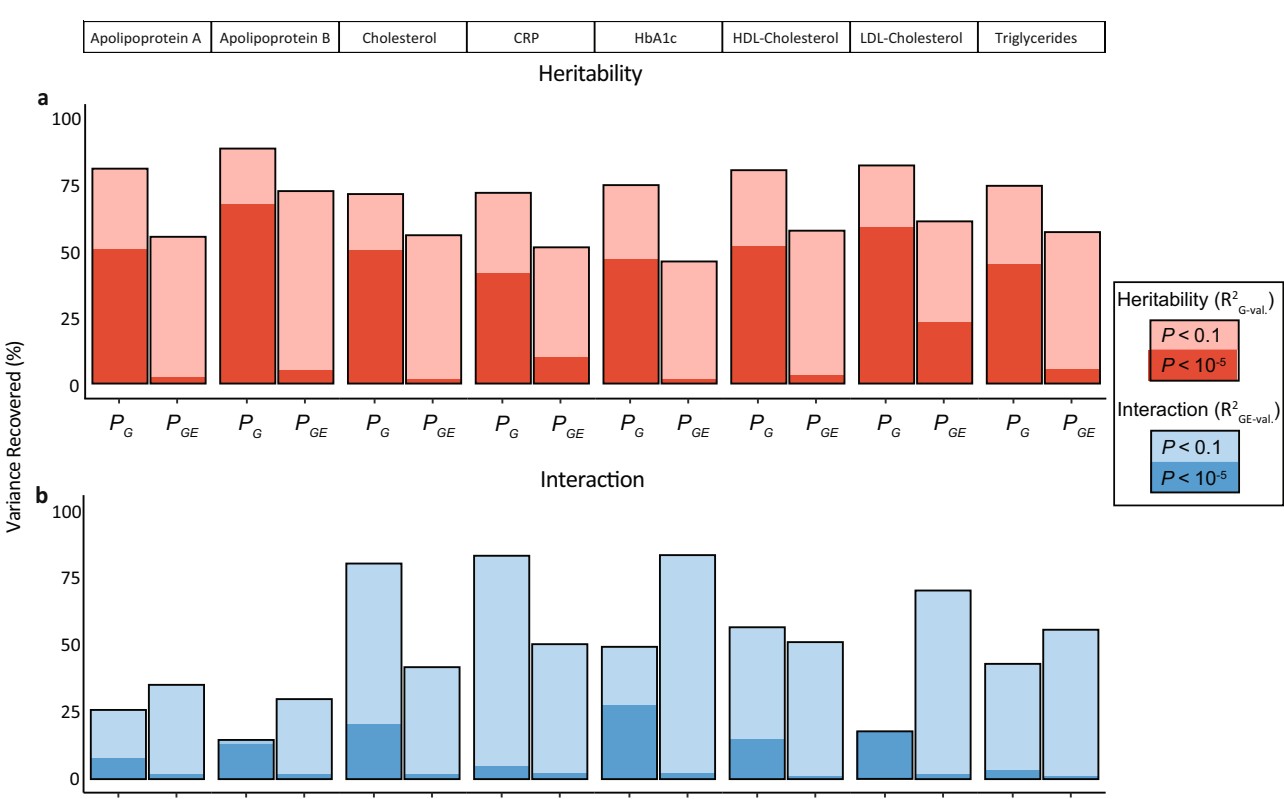

**Fig. 4 | Proportion of $R^2_{G-val.}$ and $R^2_{GE-val.}$ as a function of $P_G$ and $P_{GE}$. a** The proportion of total $R^2_{G-val.}$ recovered in the validation set at discovery sample $P_G < 10^{-5}$, $P_{GE} < 10^{-5}$, $P_G < 10^{-1}$, and $P_{GE} < 10^{-1}$ for the eight outcomes with significant interaction variance. **b** The proportion of total interaction $R^2_{GE-val.}$ recovered in the validation set at discovery sample $P_G < 10^{-5}$, $P_{GE} < 10^{-5}$, $P_G < 10^{-1}$, and $P_{GE} < 10^{-1}$ for the same outcomes. Percentages represent the proportion of variance recovered with regressors built from labelled association predictors compared to regressors with all SNPs. MonsterLM estimates in the validation set were conducted with 65,198 individuals and 1,030,579 SNPs after quality control. Source data are provided as a Source Data file.

the outcome with the highest interaction variance, CRP (Supplementary Fig. 6), with the relative increase in prediction $R^2$ ranging from 0% to 1.3% across the outcomes analyzed (for interaction significance thresholds of $10^{-4}$, $10^{-5}$; Supplementary Data 1). The largest increases in polygenic score predictiveness tended to occur in outcomes with the largest GxE variance observed (Fig. 3; Supplementary Fig. 6).

## Discussion

In this report, we developed a method, MonsterLM, to estimate variance explained by genome-wide interactions with environmental exposures. Using simulations, we verified that MonsterLM estimates the variance explained by interaction effects accurately and precisely. Analysis of UK Biobank data demonstrated the presence of significant GxE effects with WHR, a marker of metabolically deleterious adiposity. The interaction estimates for 8 of the 13 outcomes analysed were significant, ranging from 0.01 to 0.07 of overall variance, prompting further analyses into these results. The presence of significant GxE was further supported by the recovery of GxE with an exercise exposure, M10. MonsterLM was also successfully applied to dichotomous outcomes and exposures through simulations and real data. Together, real and simulated data analyses demonstrate the robustness of MonsterLM against biases such as collider effects or LD, and validates its utility as a versatile, fast and unbiased method for estimation of gene-by-environment interaction effects on biobank-scale datasets.

In benchmarking analyses, MonsterLM heritability estimates were consistent with alternative state-of-the-art methods (Table 3). When comparing MonsterLM GxE estimates to those of mtg2, MonsterLM

tends to be conservative but provides accurate and consistent estimations across simulations and plausible estimates in real data. While mtg2 is also accurate and precise in most instances, it can be limited by computational burden, consistency in simulation estimates, and plausibility for some real data estimates. When comparing MonsterLM GxE estimates to LDSC GxE, MonsterLM estimated 8 of 11 outcomes to be non-null and LDSC GxE estimated 7 of 11 outcomes to be non-null. LDSC GxE estimates ranged from 0–1.8% and were lower than MonsterLM for each outcome (as was consistent with LDSC heritability estimates versus MonsterLM, Bolt, and GRE). Some LDSC GxE advantages include the fast computational speed of summary-level statistics compared to individual-level data and robustness to stratification and common environmental effects. However, the potential for LDSC underestimation is a discussed limitation in the literature. For example, Evans et al., 2018[12] conducted a heritability model comparison study where they showed a limitation of LD score regression was its potential to underestimate $h^2$ if the trait is not highly polygenic (such as in the case of total bilirubin; Table 3). Furthermore, consistently smaller LDSC $h^2$ estimates have been shown when compared to GREML in the same data set[26] and as the lowest estimate in a recent protocols study compared to 10 other approaches (including GREML, LDAK, threshold GRMs, and SumHer)[27].

When extending the comparison to the state-of-the-art, MonsterLM provides distinct advantages over current methods for GxE analysis (Table 4)[28–36]. Several key advantages compared to other methods include: (i) computational efficiency (with two options in *CPULS* and *GPULS*) with biobank-scale individual-level data, (ii) no

model specification beyond using additive genetic coding, (iii) extensibility to dichotomous exposures and outcomes, and (iv) and demonstrated genome-wide robustness. In many settings, inference methods for genome-wide SNP-heritability and GxE make assumptions about genetic architecture. These assumptions are parametrized by polygenicity (the number of variants with effects) and MAF/LD-dependence (the coupling of effects with MAF, LD or other functional annotations). Since the true genetic architecture of any given trait is unknown, existing heritability methods may yield vastly different estimates even when applied to the same data[10–12]. This is also the case for the estimation of genome-wide environment interactions, where different assumptions about the structure of interactions result in a variety of different estimates[29–33]. Although multi-component methods that stratify SNPs by LD/MAF can address these robustness issues, fitting multiple variance components to biobank-scale data is highly resource intensive[37], and this problem is compounded when considering interactions where the number of variables analyzed increases two-fold. Alternate methods that explicitly model these dependencies are also sensitive to model misspecification[9–13]. Conversely, MonsterLM assumes an additive genetic model and does not apply further parametrization for underlying assumptions.

Significant GxE with WHR was observed for 8 of the 13 outcomes studied. Interaction effects with WHR ranged from 0.009 to 0.071, and in all cases were of smaller magnitude than their heritability counterparts. These results have important implications for future research. First, our observations suggest that GxE can contribute significantly to complex trait variance. Second, genetic associations are likely to be heterogenous when comparing populations with dramatically different obesogenic environmental exposures. The observation that a majority of GxE effects do not come from SNPs with strong marginal effects suggests this may not impact top GWAS hits excessively. We also observed the presence of significant directionality effects for marginal effect SNPs and their associated interaction effects, which suggest an overall greater impact of genetic variation under certain environmental conditions. There are also potential clinical implications for these observations. For instance, CRP reflects low-grade inflammation and is strongly associated with risk of CVD[38]. Our results suggest that genetic determinants of low-grade inflammation are dependent on adiposity distribution (WHR) and further research will be needed to understand the implications for CVD risk.

Our results also provide some further insights into why identification of GxE has been challenging. Many prior studies have reasonably focused the search for GxE on variants with genome-wide significant marginal effects. Our results show that a majority of GxE effects are due to variants with unremarkable marginal effects (i.e., only 3–28% of GxE variance recovered by SNPs with strong marginal effects), although variants with strong marginal effects remain preferred candidates for GxE interactions. We also show in a proof-of-concept analysis that incorporation of GxE can improve PS prediction, albeit very modestly.

Some limitations are worth mentioning. First, we quantile normalize all traits before analysis, and while this protects against potential scaling effects and is robust to nonnormal-distribution types, it could also bias results towards the null[39]. Second, in the event of collider effects with a covariate that is heritable through additive and heterogenous elements there could be some inflation of heritability estimates (Fig. 2; scenario 12). However, the conditions for this scenario are presumed to be quite extreme and did not affect GxE estimates. Furthermore, GxE estimates have been shown to be stable when modelling collider biases whereas genetic estimates are less well-controlled[40]. Third, information may be lost through LD pruning and from filtering rare and low frequency variants (MAF < 5%). Fourth, MonsterLM is susceptible to overestimating GxE variance when participant phenotypic and exposure missingness co-occurs in the same individuals. Fifth, the liability scale transformation for dichotomous outcomes could bias GxE estimates under specific conditions such as

violation of the normality assumption needed for the Robertson transformation (i.e. can occur in really large interaction estimates), if substantial non-additive effects exists[34], or biases towards the null due to information loss in the transformation.

In this report, we have developed a robust and well-controlled method for genome-wide GxE estimation. We established the presence of GxE in cardiometabolic traits. We observed that SNPs with weak marginal and interaction effects contribute to the majority of GxE variance. MonsterLM makes minimal assumptions about genetic architecture and is well-powered for both continuous and dichotomous outcomes. It is computationally efficient, robust, and versatile and can be used as the basis for future analyses of genome-wide environment interactions.

## Methods
### UK Biobank
The UK Biobank is a large population-based study which includes over 500,000 participants living in the United Kingdom[41,42] (https://www.ukbiobank.ac.uk/). Men and women aged 40–69 years were recruited between 2006 and 2010, and extensive phenotypic and genotypic data were collected. Quality control of genotype data was applied for individual and SNP inclusion using PLINK version 1.9 (https://www.cog-genomics.org/plink2/). We selected 325,989 unrelated British individuals (the largest unrelated cohort; 54% female and 46% male) from the UK Biobank with both genotype and trait data for inclusion in the analysis. An unrelated set of individuals were chosen to reduce genomic prediction inaccuracies[43]. Individual exclusion criteria included: (1) non-white British ancestry, (2) high ancestry-specific heterozygosity, (3) high genotype missingness (>0.05), (3) mismatching genetic ancestry, (4) sex chromosome aneuploidy, (5) mismatching gender sex and genetic sex, and (6) consent withdrawal at the time of analysis. Variants from the release version 3 of the UK Biobank data were used, which included those present in the Haplotype Reference Consortium and 1000 Genomes panels with imputation quality > 0.7 and had no deviation from Hardy-Weinberg equilibrium ($P > 1 \times 10^{-10}$)[42]. Our study focussed only on common variants; thus, genotypes were filtered by removing highly correlated SNPs with an LD $r^2 > 0.9$ and removing SNPs with a MAF < 0.05. SNP exclusion criteria included: (1) SNPs with low imputation quality (INFO score ≤0.30), (2) call rate <0.95, and (3) ambiguous or duplicated SNPs. After all quality control (QC) filters, 1,030,579 SNPs and 325,989 individuals remained. Genetic variants were partitioned to minimize the number of blocks on each chromosome, with each block having a maximum SNP count of 25,000. Genotypes were standardized to have a mean of zero and standard deviation of one. We examined eleven continuous traits and two (dichotomous) cardiometabolic outcomes including: Apolipoprotein A, Apolipoprotein B, Total Cholesterol, C-reactive protein (CRP), Glucose, HbA1c, HDL-Cholesterol, LDL-Cholesterol, Triglycerides, Total Bilirubin, height, coronary artery disease (CAD) and Type 2 Diabetes (T2D). WHR, an exercise parameter, smoking status, and a randomly generated exposure were used as environmental exposures ($E$) to quantify GxE interactions.

For follow-up analyses, and to avoid the potential for overfitting from sample overlap[44], we randomly partitioned the UK Biobank participants into two sets: a discovery set containing 80% of the participants used for model building and a validation set containing the remaining 20% of the participants.

### MonsterLM estimations of variance explained by GxE effects
MonsterLM estimates heritability (G) and GxE effects in three steps outlined in Fig. 5. This can be performed using three variable-type combinations: continuous exposures and outcomes, dichotomous exposures and continuous outcomes, and continuous exposures and dichotomous outcomes. Slightly different configurations of MonsterLM are used depending on the variable-type combination. Step one

**Table 4 | Comparison of current methods estimating gene-by-environment contributions to MonsterLM**

| Method | Description | Advantages | Limitations | MonsterLM |
|---|---|---|---|---|
| StructLMM[29] | Evaluates interaction variance for multiple environmental factors with a single SNP. | Fast, robust model for a variety of different environmental exposures. | Limited to interaction effects of only a single SNP or genotype. | Analyzes variance explained by interactions genome-wide (after LD-pruning). |
| CGI-GREML[30] | Uses a mix of parametrized models and restricted likelihood methods to estimate GxE. | Well-structured for identifying GxE interactions with categorial exposures. | >250 Likelihood Ratio Tests; Slow; Cannot use continuous traits. | Can analyze continuous traits without categorizing them; quick, efficient Wald-test for $R^2$. |
| GxEMM[31] | Linear mixed model method to detect GxE interactions across the genome and a single exposure. | Multiple parametrizations available to efficiently model GxE interaction effects. | Small sample size only; Minimal number of SNPs. | Can analyze a large sample size with many SNPs via genotype partitioning and conjugate gradient method. |
| GRSxE[32] | Method to detect total GxE interactions with a Gene-Risk Score. | Estimates the GxE contribution for all possible environmental factors with SNPs. | Assumes each SNP interacts equally with E. | Accounts for the unique interaction effect of each SNP with E. |
| LEMMA[33] | Linear mixed model method to detect GxE interactions across the genome and an estimated linear combination of exposures. | Considers the impact of over-lapping environmental exposures when computing total GxE contributions across the genome. | Requires parametrization and model specification; uses an estimated linear combination of exposures, assuming all E's interact with the same SNPs | Tests for specific interaction with E rather than a linear combination of E. |
| GxESum[34] | Estimates genome-wide GxE variance using GWAS summary statistics. | Has controlled type I error rates and unbiased GxE estimates; efficient computation-wise; can also be applied to binary disease traits; summary statistics advantages. | Disadvantages of using summary statistics include information limitations and population stratification bias susceptibility. | Individual-level data advantages include better handling of LD and complex interactions. |
| LDSC GxE[35] | Estimates genome-wide GxE variance using GxE GWAS (GWIS) summary statistics. GWIS replaces GWAS in standard LDSC regression[37] to estimate $h^2_{GxE}$. | Utilizes GWAS summary statistics; computationally efficient; typically robust to confounding from stratification and common environmental effects. | Risk of $h^2$ underestimation in high LD or low polygenicity regions; bias when LD scores from reference population and GWAS mismatch. | Individual-level data advantages include better handling of LD and complex interactions; robust in low polygenicity scenarios. |
| MTG2 IGE[28,36] | Estimates variances explained by additive effects of exposure variables, by ExE interactions, and covariance between genetic effects and exposomic effects. | Precise estimates with low standard deviations, flexible for a variety of exposure-based interactions. | Requires generation of a genomic relationship matrix file (grm file) using PLINK which requires >1TB RAM as N > 100,000. | Does not require generating a genomic relationship matrix, is computationally efficient, and can use biobank-scale individual level data. |

Existing methods of biobank GxE estimation compared to MonsterLM. A description of eight methods to estimate GxE estimates with respective advantages, limitations, and comparisons to MonsterLM.

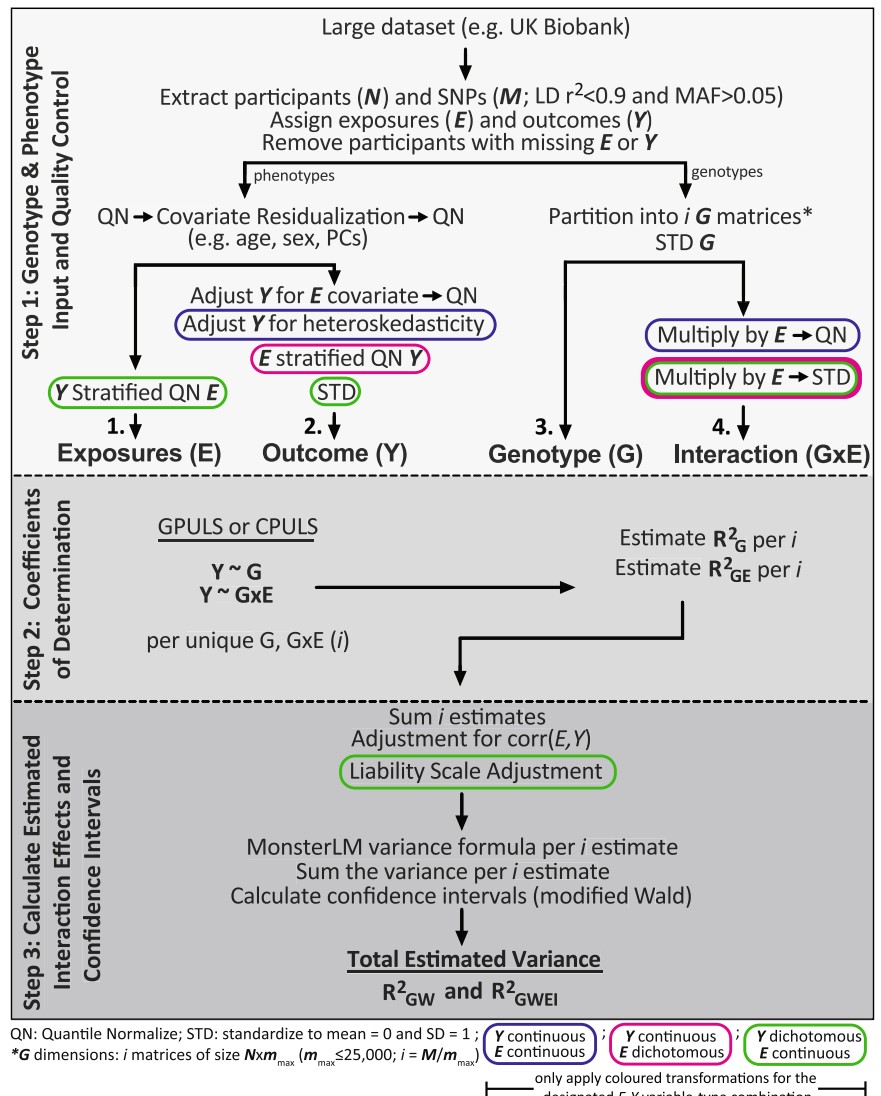

**Fig. 5 | The MonsterLM method split into three steps for continuous outcomes and exposures.** The first step describes data processing, the second step describes methods of computing least squares, and the third step describes how to finalize estimates and compute confidence intervals. Sections outlined by blue, red, or green are transformations only to be applied with that variable-type combination described in the footnote. $E$: exposure matrix; $Y$: outcome matrix; $G$: genotype matrix; $GxE$: interaction matrix. $N$; number of participants; $M$; number of SNPs; $m_{max}$: maximum number of SNPs to be partitioned genotype matrix.

processes exposure, outcome, interaction, and genotype data; step two calculates the coefficients of determination ($R^2$); and step three calculates the estimated G and GxE with confidence intervals.

## Step one: genotype and phenotype input and quality control

The standard linear model for an outcome, $Y$, when an interaction term is included can be expressed as:

$$Y = \beta_G G + \beta_E E + \beta_{GE} GE + \epsilon \tag{1}$$

Where $G$ is the standardized genotype matrix, $E$ is the quantile normalized environmental exposure, $GE$ is the product between each genotype matrix and environmental exposure, resulting in a matrix with the same dimensionality as $G$. $G$ is coded in the additive model ({0,1,2}) and standardized so that the mean = 0 and standard deviation = 1 for each SNP. $GE$ is the quantile normalized product of $G$ and $E$. The betas (β) represent the true marginal effects associated with their respective term and $\epsilon$ represents the random error term. In practice, participants are selected to avoid missing data in $Y$ and $E$. Both $Y$ and $E$ are first quantile normalized, then regressed for age, sex, and population stratification (first twenty genetic principal components), and

quantile normalized once more. Two more transformations are applied to $Y$: (i) regression of the processed $E$ then quantile normalization; (ii) and a heteroscedasticity adjustment for continuous outcomes. The aforementioned processing assumes continuous $Y$ and $E$ variables.

## Step two: calculating the coefficients of determination

We denote the matrix of $G$ or $GE$ as $U$ with dimensions $n \times m$, where $n$ is number of participants and $m$ is number of SNPs:

$$U = G, \text{ or } U = GE \tag{2}$$

As the environmental exposure is residualized from $Y$, we can leave $E$ out of the model.

The standard linear model becomes:

$$Y = \beta_U U \tag{3}$$

Given $Y$, the least squares estimate for $\hat{\beta}_U$ is:

$$\hat{\beta}_U = \left( U^T U \right)^{-1} U^T Y \tag{4}$$

After computing $\hat{\beta}_{U_i}$, the predicted values of $Y$ denoted as $\hat{y}$, are given by:

$$\hat{y} = \hat{\beta}_U U \qquad (5)$$

MonsterLM enables multiple linear regression on biobank-scale datasets by parallelizing the calculation of least squares regression. The calculation is done such that the only practical limitation is the speed of the inversion of the $U^T U$ matrix, without any restriction on $n$. This limitation is circumvented using the conjugate gradient method and GPU acceleration (henceforth referred as GPULS; Supplementary Table 8 [21]). If users are constrained by GPU hardware but have adequate RAM allocation (>200 GB RAM) then a CPU-least squares method (henceforth referred as CPULS) can be used to compute traditional least squares in parallel to estimate block-wise $R^2$ (Fig. 5; Step 2). MonsterLM assumes an additive genetic model but does not make further assumptions regarding genetic architecture (such as polygenicity of effects, MAF and LD). Genotypes are partitioned into blocks with a maximal size of 25,000 SNPs ($m$) to minimize LD spillage between blocks and to optimize speed of the matrix calculation.

### Step three: calculating total genetic and interaction estimates and confidence intervals

Once $\hat{y}$ is calculated for each block $i$ using either $G$ or $GE$, both $R^2$ and adjusted $R^2$ can be derived for additive genetic effects and interaction effects, respectively. The total genome-wide contribution of additive genetic effects ($R^2_{GW}$) and GxE interaction effects ($R^2_{GWEI}$) is given by summing adjusted $R^2$ over all blocks:

$$R^2_{GW} = (1 - R^2_{E,Y}) \sum_{i=1}^{j} R^2_{G_i} \qquad (6)$$

$$R^2_{GWEI} = (1 - R^2_{E,Y}) \sum_{i=1}^{j} R^2_{GE_i} \qquad (7)$$

Where $j$ is the number of blocks used per analysis (i.e. 60 blocks for current analyses) and $1 - R^2_{E,Y}$ is an adjustment to account for the fact that $Y$ is residualized for $E$. $R^2_{E,Y}$ is the coefficient of determination of $Y$ and $E$.

$R^2_{U_i}$ is the adjusted $R^2$ per block for $G$ or $GE$, with $n$ the sample size and $m$ the number of predictors. The 95% confidence (CI) of $R^2_{U_i}$ can be estimated for each block by first calculating the variance of the squared multiple correlation coefficient using Kendall and Stuart's method of variance estimation[45] available as "Variance.R2" in the MBESS[46] R package where:

$$\widehat{Var}_{\infty}\left(R^2_{U_i}\right) = \left(1 - R^2_{E,Y}\right)\left(\frac{n-1}{n-m-1}\right)^2 \frac{(n-m-1)(n-m+1)}{(n^2-1)} HyperG(R^2_{U_i}, n, m) \qquad (8)$$

and $HyperG(R^2_{U_i}, n, m)$ is the asymptotic adjustment using the hypergeometric function discussed in section 2B of Stuart, Ord, and Arnold (1999)[45]. The 95% CI for a single block, $i = 1$, can then be derived using the Wald estimate:

$$95\% \, CI = R^2_{U_i} \pm 1.96 \sqrt{\widehat{Var}\left(R^2_{U_i}\right)} \qquad (9)$$

To estimate the 95% CI for our genome-wide $G$ or $GxE$ estimate, $R^2_{GWU}$, we calculate the total asymptotic variance as the sum of the individual variances ($R^2_{U_i}$) for $j$ blocks:

$$\widehat{Var}\left(R^2_{GWU}\right) = \sum_{i=1}^{j} \widehat{Var}(R^2_{U_i}) \qquad (10)$$

where $\widehat{Var}\left(R^2_{U_i}\right)$ is each $i$ variance estimate from Eq. (8). For the total asymptotic variance estimated, we calculate the 95% CI of $R^2_{GWU}$ as:

$$95\% \, CI = R^2_{GWU} \pm 1.96 \sqrt{\widehat{Var}\left(R^2_{GWU}\right)} \qquad (11)$$

### MonsterLM for dichotomous outcomes and exposures

Applying MonsterLM with dichotomous exposures and continuous outcomes uses the same algorithm as with continuous variables (Fig. 5) with a few key modifications. These include: (i) no exposure modification (Step 1: $E$), (ii) the continuous outcome is quantile normalized in each dichotomous group separately (referred to as "$E$ stratified QN $Y$") in Step 1, and (iii) standardizing ($\mu = 0$, $\sigma = 1$) the interaction ($GxE$) terms (Step 1).

MonsterLM can also be applied to dichotomous outcome variables and continuous exposures (Fig. 5) with the following modifications: (i) the continuous exposure in each dichotomous outcome group is quantile normalized separately (referred to as "$Y$ stratified QN $E$"); (ii) standardizing ($\mu = 0$, $\sigma = 1$) the dichotomous outcomes (Step 1; $Y$), and (iii) and applying a liability scale transformation[47] on the total estimates (Step 3) (Fig. 5).

### Validation of MonsterLM using simulations

MonsterLM was tested using simulations under a range of scenarios. In all simulations, "real" genotypes from 325,989 UK Biobank participants (as described below) were used and outcomes and exposures were simulated. Unless otherwise stated, outcomes and exposures were simulated assuming true (unobserved) effects ($\beta_G$, $\beta_E$, $\beta_{GE}$) following a standard normal distribution. 20% of SNPs were randomly selected to have a marginal effect on the simulated trait of interest, $Y_{sim}$ (i.e. $\beta_G \neq 0$). We further assumed that 2% of total SNPs have an interaction effect (i.e. $\beta_{GE} \neq 0$). The error ($\epsilon$) was sampled from an independent and identically distributed standard normal distribution. The simulated trait ($Y_{sim}$) and simulated exposure ($E_{sim}$) were then computed as:

$$Y_{sim} = \beta_G G + \beta_E E_{sim} + \beta_{GE} GE_{sim} + \epsilon \qquad (12)$$

We tested 12 scenario conditions through simulations (Fig. 1; top panel). Two model types, "base" and "collider" were considered. Firstly, base model simulations considered that $E$ was not dependent on $G$ (i.e. $E$ is not heritable) and the genetic and interaction effects for all SNPs were randomly generated from a standard normal distribution. Secondly, we considered models including a collider bias. A collider bias can occur when two conditions are met: a controlled-for environmental exposure is both heritable and influenced through an unobserved confounder; and that same unobserved confounder influences the outcome (Fig. 1; top panel). As shown in Aschard et al. 2015[48] and Akimova et al., 2021[40], a collider bias can potentially lead to overestimation of GxE variance and other variance components. Collider model simulations had a set correlation R² between $Y_{sim}$ and $E_{sim}$ of 0.2 and assumed that the correlation was entirely due to the simulated unobserved confounder covariate ($UC_{sim}$), an extreme collider model (Fig. 1; top panel). In these scenarios, $UC_{sim}$ was simulated to have 20% of its variance explained by additive genetic effects (i.e. heritability of $E_{sim}$), as WHR was observed to have similar heritability empirically. In one collider scenario, genetic heterogeneity (which has been explored in the context of GxE models[49]), explained 20% of the genetic variance of $E_{sim}$ (with the remaining genetic variance of $E_{sim}$ explained by additive genetic effects). Genetic heterogeneity was simulated by

including a genetic interaction with a randomly generated binary variable. Genome-wide simulated variances were set at $R^2_{GW} = 0.20$ (i.e. heritability of $Y$), and $R^2_{GWEI} = 0.1$ or $0.0$ (i.e. interaction variance of $Y$). In base model simulations, observed correlations between $E$ and $Y$ are due to the true set value (i.e. causal effect of $E$ on $Y$). In collider simulations, there is no causal effect of $E$ on $Y$ or $Y$ on $E$ such that any observed correlation is entirely due to the collider bias.

For each simulated scenario, ten simulations were performed (Fig. 2; legend). Scenarios 1–8 use the base model format and scenarios 9–12 use the collider model format. The complete set of scenarios include: (i) null GxE effect ($R^2_{GWEI} = 0.0$); (ii) non-null GxE effect ($R^2_{GWEI} = 0.1$); (iii) non-null GxE effect with non-null exposure variance ($R^2_{E\ on\ Y} = 0.1$); (iv) non-null GxE effect where all $\beta_{GE}$ effects are sampled from SNPs in the highest LDscore[37] quartile; (v) non-null GxE effect where all $\beta_{GE}$ effects are sampled from SNPs in both the highest LDscore quartile and lowest MAF quartile; (vi-vii) non-null GxE effect when sampled $\beta_{GE}$ effects have exponential and beta distributions (positive kurtosis); (viii) non-null GxE effect with a dichotomous exposure; (ix) non-null GxE effects in a collider scenario; (x) non-null GxE effect in a collider scenario where $\beta_{GE}$ effect SNPs were not an element (completely non-overlapping) of $\beta_G$ effect SNPs; (xi) null GxE effect in a collider scenario where $\beta_{GE}$ effect SNPs are a strict subset (completely overlapping) of $\beta_G$ effect SNPs; and (xii) null GxE effect in a collider scenario where $E_{sim}$ is heritable through additive and heterogenous genetic effects.

MonsterLM robustness was further investigated by testing the impact of mean imputation for both the $E_{sim}$ and $Y_{sim}$. We examined scenarios with mean imputation of $E_{sim}$ only, $Y_{sim}$ only, and $E_{sim}$ and $Y_{sim}$ within the same participants to further address any biases in model design. The previously described scenario 2 conditions were used for these latter simulations.

MonsterLM performance was also validated with dichotomous outcomes in scenario 2 conditions. Genome-wide simulated variances were set at $R^2_{GW} = 0.20$ (i.e. heritability of $Y$), and $R^2_{GWEI} = 0.0$ (i.e. interaction variance of $Y$) to assess for type I error.

## Applying MonsterLM to UK Biobank traits and exposures

The MonsterLM method was applied to 13 outcomes from the UK Biobank in 325,989 unrelated British participants. The tested outcomes were clinically pertinent blood biomarkers, major diseases, and height (a well-studied heritable outcome). MonsterLM was applied as outlined in Fig. 5. Four different exposures were tested in total: WHR, days of at least 10 minutes moderate weekly physical activity status (M10), smoking status, and a randomly permuted exposure. WHR was selected as an environmental exposure because it is a measure of central obesity linked to a wide range of adverse metabolic consequences, including diabetes and cardiovascular disease (CVD)[50]. M10 was selected due to its relevance as an obesogenic risk factor but minimal to negligible estimated additive genetic variance and correlation with outcomes ($R^2_{G,E} = 0.02$, $R^2_{E,Y}$ range from 0.00 to 0.01). This serves as a suitable control to reduce the possibility that any spurious collider effects or additive genetic effects would explain any of the interaction variance. Smoking status was chosen as a dichotomous exposure[51,52]. Lastly, a permuted exposure was added as a negative control of no interaction.

## Directionality of effects analysis

After computing $R^2_{GW}$ and $R^2_{GWEI}$ for our 13 outcomes, we tested whether the direction of effects was concordant between marginal and interaction regression coefficients for each SNP in the significant eight outcomes. Concordant direction of effects is defined as when $\hat{\beta}_G$ has the same sign (+/+, −/−) as $\hat{\beta}_{GE}$ for a single SNP and its associated interaction. Discordant direction of effects is defined as when the $\hat{\beta}_G$ and $\hat{\beta}_{GE}$ have a different sign (+/−, −/+) for a single SNP and its associated interaction. We used a subset of $\hat{\beta}_G$ and $\hat{\beta}_{GE}$ coefficients that were in low LD ($r^2 < 0.1$) and computed the direction of effect

concordance for this subset. We then plotted the sign concordance twice: first as a function of univariate $\hat{\beta}_G$ $p$-values ($P_G$), then as a function of univariate $\hat{\beta}_{GE}$ $p$-values ($P_{GE}$), which were computed from association of single SNPs and their respective interaction on the outcomes. Two-proportion $Z$-tests were used to compare the proportion of directionally concordant marginal and interaction effects for each outcome in each threshold compared to a null proportion of 0.50.

## Stratification of estimates by MAF and LD

SNPs were stratified by MAF and LDscore into a total of twenty bins: five MAF bins ($0.05 \leq 0.1$, $0.1 < MAF \leq 0.2$, $0.2 < MAF \leq 0.3$, $0.3 < MAF \leq 0.4$, and $0.4 < MAF \leq 0.5$) and four LDscore quantiles ($0 < LD \leq 0.25$, $0.25 < LD \leq 0.50$, $0.50 < LD \leq 0.75$, and $0.75 < LD \leq 0.9$). MAF and LDscore were calculated using a subset of 5000 participants from the UK Biobank. We then computed the variance explained ($R^2_{GW}, R^2_{GWEI}$) and divided each estimate by the total number of SNPs in each bin to get an $R^2$ per SNP value that was compared between bins and to the total genetic and interaction variance estimates.

## Polygenic scores analysis

We calculated polygenic scores (PS) without interactions ($PS_G$) for outcomes with statistically significant GxE variance. We first selected SNPs based on the univariate association $p$-value from regression of each variant with outcomes from the discovery set (randomly chosen 80% of UK Biobank sample). We then combined the selected SNPs into a single block from the discovery set and applied MonsterLM regression to obtain the multiple linear regression coefficients ($\hat{\beta}_G$). Using these coefficients, we calculated the $PS_G$ in the validation set as:

$$PS_{G,i} = \sum_{j}^{O} G_{i,j} \hat{\beta}_{G,j} \tag{13}$$

Where $PS_{G,i}$ is the individual polygenic score of participant $i$, $j$ is the SNP number and $O$ represents the total number of SNPs included in this analysis. We then evaluated the predictiveness of each $PS_G$ using $R^2$ in the validation set (remaining 20% of the UKB sample). We repeated the same process for four univariate $P_G$ thresholds ($10^{-2}, 10^{-3}, 10^{-4}, 10^{-5}$) for each outcome.

We define $PS_{GE}$ as the $PS$ with GxE interactions included. To include GxE interactions, we selected significant interactions based on the univariate association $p$-value from regressing each GxE interaction with outcome concentration in the discovery set. These interactions are selected from the subset of SNPs included in polygenic scores without interactions. The interactions passing the univariate $P_{GE}$ thresholds ($10^{-2}, 10^{-3}, 10^{-4}, 10^{-5}$) were then included with the SNPs to create a single block. We applied MonsterLM regression to obtain the multiple linear regression coefficients ($\hat{\beta}_G$, $\hat{\beta}_{GE}$). Using these coefficients, we calculate the $PS_{GE}$ as:

$$PS_{GE,i} = \sum_{j}^{O} G_{i,j} \hat{\beta}_{G,j} + \sum_{k}^{P} (GxE)_{i,k} \hat{\beta}_{GE,k} \tag{14}$$

Where $PS_{GE,i}$ is the polygenic score with interactions incorporated for participant $i$, summed over each SNP ($j$) and, if included, its associated interaction ($k$). $O$ represents the SNPs included in the $PS_{GE}$, while $P$ represents the interactions included, a subset of $O$. As with the $PS_G$, we evaluated the predictiveness of each polygenic score using $R^2$ in the validation set. We repeated for all pairwise combinations of the four $P_G$ thresholds and the four $P_{GE}$ thresholds, resulting in 16 $PS_{GE}$ for each outcome in addition to four $PS_G$.

## Comparison with other methods

We compared MonsterLM estimates to alternative methods. For heritability, MonsterLM estimates were compared to BOLT[53] (https://

alkesgroup.broadinstitute.org/BOLT-LMM/BOLT-LMM_manual.html)
and GRE[17] (https://github.com/bogdanlab/h2-GRE). MonsterLM herit-
ability and GxE estimates were compared with mtg2 IGE[28] (https://bio.
tools/mtg2) and LDSC[37] (https://github.com/bulik/ldsc).

### Power calculations

Statistical power was estimated using sets of 10,000 simulations. Non-
central F-distributions were used to simulate the observed genetic and
interaction effects at each genotype block, and genome-wide herit-
ability and GxE estimates derived as previously described. True set
adjusted $R^2$ ranged from 0.05 to 0.5. Sample size ranged from
$N = 50,000$ to 400,000 individuals by increments of 10,000. For each
condition, power was defined as the proportion of observed $p$-values
less than 0.05 out of the 10,000 simulations.

### System requirements

MonsterLM software (Supplementary Software 1) can be run on all
major platforms (e.g. GNU/Linux, macOS, Windows). For biobank-scale
analyses, recommended hardware requirements are a unix-like virtual
environment supporting a minimum of 250 GB RAM space for in-
memory operations. System GPUs are optional and can be used to
speed up matrix inversion. Software requirements include the pro-
gram dependencies: BASH (≥5.0), R (≥3.6.3), and GPULS (optional).
Essential R dependencies include the packages: tidyverse, data.table,
MBESS, and gsl.

### Reporting summary

Further information on research design is available in the Nature
Portfolio Reporting Summary linked to this article.

## Data availability

This research has been conducted using individual genetic and phe-
notypic data obtained from the UK Biobank (http://www.ukbiobank.ac.
uk/), under application #15255. The UK Biobank study received
approval from the National Health Service National Research Ethics
Service North West. Access to the UK Biobank individual-level data is
not publicly available and must be obtained via an application (https://
www.ukbiobank.ac.uk/register-apply/). All other data supporting the
findings described in this manuscript are available in the article and its
Supplementary Information files. Source data are provided with
this paper.

## Code availability

The software package containing all code and relevant documentation
to run MonsterLM is available in a public GitHub repository at https://
github.com/GMELab/MonsterLM (Supplementary Software 1; https://
zenodo.org/record/8092995)[54].

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

## Acknowledgements

The authors are thankful for all the UK Biobank participants. We would also like to thank Dr. Andrew McArthur and his laboratory at McMaster University for providing GPU support. M.D. is supported by a Canadian Institute of Health Research Doctoral Award, a Mach-Gaensslen Foundation Award, and a McMaster Undergraduate Medical Education Research scholarship.

## Author contributions

M.D.: data curation, software, formal analysis, investigation, visualization, writing; M.K.: data curation, software, formal analysis, investigation, visualization, writing; S.M.: data curation, software, formal analysis; M.C.: data curation, analysis interpretation, writing (review and editing); C.J.: formal analysis, visualization, writing (review and editing); Na.P.: formal analysis; Ni.P.: formal analysis; WN: software; R.L.: software; S.D.: software; R.M.: visualization, writing (review and editing); J.P.: software; G.P.: conceptualization, supervision, funding acquisition, methodology, project administration, writing (review and edit).

## Competing interests

M.C. has received consulting fees from Bayer. G.P. has received consulting fees from Bayer, Sanofi, Amgen, and Illumina. The remaining authors declare no competing interests.
