## [Peer Review File · Nature Communications]

A versatile, fast and unbiased method for estimation of gene-by-environment interaction effects on biobank-scale datasetsREVIEWER COMMENTS

Reviewer #1 (Remarks to the Author):

A versatile, fast and unbiased method for estimation of gene-by-environment interaction effects on biobank-scale datasets

Khan et al.

This paper proposes a novel GxE method that is an extension of GRE model (#19 in references). It is basically a series of multiple regressions, each with a handful number of SNPs (# SNPs < sample size) from which R^2 values are summed to estimate genome-wide GxE variance. The authors applied the proposed method to the UK Biobank for 8 biomarkers as the main phenotypes and waist-hip-ratio (WHR) as the environmental variable. The phenotypic variance explained by estimated GxE ranged from 11 – 58%, which is surprisingly large. The authors show a significant improvement of polygenic risk score prediction when using their GxE model, compared to the null model (without GxE).

This study is interesting and important as GxE is now widely recognised as an important source of phenotypic variation. However, there are several comments and questions that may help improve the current version of the manuscript.

1. The authors should verify their novel method using a wider range of simulation scenarios. For example, the size of GxE variance (R^2) is 0.005 or 0.015 (Figure 2). Are the estimates of the proposed model still unbiased when using a large GxE variance (e.g. 0.1 or 0.2)? Note that the authors reported that ~ 50% of phenotypic variance can be explained by GxE.

Please also consider the genetic and residual correlations between the main phenotypes and environmental variable. Ni et al (2019) reported that estimated GxE variance can be biased when such correlations are not properly modelled.

Ni et al. Genotype–covariate correlation and interaction disentangled by a whole-genome multivariate reaction norm model. Nature communications 10, 2239 (2019)

The authors should also assess if the type I error rate is controlled under the null model (in the absence of GxE).

2. Can this proposed method be applied to binary disease traits? If so, simulation of binary variables should be used to verify the method. Otherwise, the authors should add a limitation to clarify this. Please also note that there are a number of existing methods (including GxEsum that can be based on GWAS summary stats) can be applied to disease traits, which should be acknowledged in Discussion.

GxEsum: genotype-by-environment interaction model based on summary statistics. bioRxiv (2020) doi: <https://doi.org/10.1101/2020.05.31.122549>

3. Can this proposed method be applied to binary or ordinal categorical environmental variables? Although the authors used WHR (continuous) as the environmental variable, a substantial proportion of environmental variables is not continuous. The authors should comment or discuss on this issue with proper references (several GxE models may allow using binary or ordinal categorical environmental variables).

4. In the simulation, the authors used a single chromosome to verify their method. The authors should use all chromosomes because the proposed model may not be able to account for correlation between chromosomes if there is any. This is also a concern for the original GRE paper (that seems using a single chromosome to verify GRE model). Even with samples after relatedness cut-off QC, correlation among chromosomes can be non-negligible (see Figure 1 in the following reference).

Estimation of genomic prediction accuracy from reference populations with varying degrees of relationship (2017) PLoS ONE 12: e0189775.

My point is that the sum of estimates across chromosomes can be overestimated unless correlations among chromosomes are properly modelled. The authors should revisit their estimates from the real data (up to 58% of phenotypic variance).

5. Line 214-220. Did the author really investigate their estimates when varying parameters that determines the genetic architecture? For example, the authors should check if the estimates are robust to different values of scale factors used in reference #9, 10 and 11 (specifically alpha in equation (1) in reference #10). Please also see supplementary Table 12 in Ni et al Nature communications 10, 2239 (2019) that shows estimated GxE variance is robust to misspecified scale factors.

6. In relation to the question above, the authors mentioned the proposed method makes no assumption with respect to the genetic (heritability) model (line 227). However, it uses standardised genotype

coefficients, i.e. scale factor (α) = -0.5 (please see equation (1) in reference #10). Would the proposed method be able to get the same results (in terms of R^2) without standardisation?

<Minor comments>

I found a number of references are not very relevant, e.g. in line 137, Please check if the reference #16 reported any estimated SNP-heritability for those traits. In line 224, is reference #16 appropriate? The authors should carefully check references and their appropriateness through the text.

Line 222. Did the studies reported that their GxE estimates were biased? Or, did the authors check and confirm that GxE estimates across those methods were significantly different from each other, using the same data? Please provide more explicit evidence to support the argument.

Line 265. '... specific GxE interactions' should be 'GxE interaction for specific variants'?

What is the reason to select 8 biomarkers as the main phenotypes? Why not for height or BMI that are well studied-model traits?

Equation (3). I don't think the vertical bar is a correct notation for this expression. Please revise.

Reviewer #2 (Remarks to the Author):

The basic idea of the manuscript is to apply the GRE method of Hou et al. 2019 to gene-environment interactions. While I think this is potentially a good idea, the authors have not convinced me that their method works as advertised.

A major concern I have is that they first residualise the phenotype for the environmental exposure before fitting the gene-environment interaction models. When the phenotype in question is highly correlated with the environmental exposure, as is likely to be the case for much of their empirical analyses, this means that the variance components they estimate are not representative of the variance components of the original phenotype before residualisation.

It appears that fitting the environmental exposure jointly with the GxE interaction model would only add 1 parameter to the model, so why not do this?

I found the simulations performed to be inadequate. The simulations only consider a single chromosome. It would be better to simulate a model based on genome-wide SNPs, and to see how the method performs when summing over chromosomes. It is also unclear exactly how the authors' method of calculating adjusted R^2 compares to the methods of Hou et al. 2019. Could the authors compare their heritability estimates to those from the method of Hou et al.? Further, Hou et al. split the genome by chromosome, which should give approximately independent blocks in a homogeneous population. Why don't the authors also split by chromosome? Also, can the authors demonstrate that their standard errors and confidence intervals are accurate?

The authors find very large interaction variance components. If true, this is an important result. However, they are larger than the heritable components for some traits, which I find somewhat implausible -- I wonder if this could be an artefact of first residualising the biomarkers for waist-hip-ratio. The authors show that most of their signal is coming from SNPs without significant marginal effects. Do these SNPs also exhibit effects on the phenotypic variance? As would be expected if they are involved in interactions.

Could the authors compare their GxE heritability estimates to those from the method of Ni et al. 2019 (Genotype-covariate correlation and interaction disentangled by a whole-genome multivariate reaction norm model)?

Finally, I worry that some of what they are observing is the result of a non-linear relationship between phenotype and environmental exposure. If these phenotypes have a non-linear relationship with WHR, and WHR is heritable, then we might expect to find what appears to be GxE interactions, but is in fact GxG interactions due to a non-linear relationship between WHR and phenotype.

Minor comments:

Line 117: it would help to explain briefly the simulation scenarios in the main text

Line 285: were any further sample quality control filters applied beyond ancestry?

Line 392: 'tested whether direction' -> 'tested whether the direction'

Methods section: The authors appear to have the effect vector left-multiplying the genotype matrix. Shouldn't it be right-multiplying?

Reviewer #3 (Remarks to the Author):

See attached word document

The authors propose a new approach to estimate the contribution of GxE to variance of quantitative outcomes in large dataset. The manuscript is well written and the positioning of the method against the state-of-the-art is clear (Table 1) and demonstrates it is fulfilling a gap in the field. The methodology behind the approach is relatively straightforward, and most of the novelty comes from the implementation, including in particular 1) using conjugate gradient and GPU to improve computational time, and 2) estimation of effect using a formal least square estimation across thousands of variants without using a penalization. However, I have major concerns regarding the application of the method using waist to hip ratio as an environmental exposure. While I understand the motivation (WHR potentially captures multiple environmental effect), I am worried using an anthropometric trait as a proxy for the environment can severely bias the estimation of both G and GxE effect. I highly recommend using a standard environmental exposure to demonstrate the performance of the method in real data. More details on this point along additional comments are provided below.

1. The marginal genetic heritabilities estimated using MonsterLM in Figure 3 (BTW --tabulated values would be useful) appear to be substantially larger than previously reported GWAS heritability (extracted from the LDhub database) with e.g. ~ 0.30 vs 0.15 for HDL, ~ 0.19 vs 0.13 for LDL, ~ 0.25 vs 0.07 for HbA1C. The authors should derive heritability estimates and standard deviation using existing methods (GCTA or LDscore or LDKA, etc) on the same data they use in the MonsterLM application and discuss potential differences.
2. As mentioned above, treating WHR as an exposure is clearly my major concern. Adjusting for a heritable covariate can induce a bias in the effect estimates of the regression coefficient between the outcome and the SNP tested (a so-called collider bias, PMID= 25640676). The impact of the bias would depend on the phenotypic correlation between WHR and the outcome considered, and the heritabilities and co-heritabilities. It is unclear to me to which extent this can impact the estimation of GxE and the follow-up analyses conducted by the authors (distribution of GxE across variants, polygenic risk prediction, etc). Here are a few takes on this:
 - a. According to the LDhub database, several of the outcome studied have substantial and sometimes highly significant genetic correlation with WHR (e.g. -0.525 , $P=10^{-12}$ with HDL, and 0.426 , $P= 2.2e-16$ with Triglyceride), thus potentially modifying the effect of variants marginally associated with HDL and triglyceride. Conversely, modelling a GxWHR on outcomes with small genetic correlation with WHR can in principle induce genetic effect at variants with no marginal effect on the primary outcome. It would be wise to check potential correlation between the genetic effect on WHR and the marginal genetic and interaction effect estimated for each outcome considered.
 - b. The simulation using G-E correlation (scenarios 2 & 3) is a commendable step toward quantifying the potential impact of using exposure with a genetic component. However, there is subtle, nonetheless important alternative model to consider that likely better match the use of WHR: when the E has no causal effect on Y, but instead simply share phenotypic variance. Typically $E = U + G1 + e1$ and $Y = U + G2 + e2$, where U is a non-genetic shared risk factor, G1 and G2 are (correlated or not) genetic components, and e1 & e2 are independent residuals variances. A toy R simulation code is provided at the end of the review. As shown in this example, using an exposure not causal to Y but simply co-varying with it can induce bias on both G and GxE. For the GxE bias I arbitrarily used a hypothetical GxUnmeasured factor (S) interaction on WHR to illustrate my point. A likely more realistic scenario –and my best guess (but more painful to check through simulation)– is that correlation between tag SNPs and causal ones induces similar bias. While such bias would be modest, it could be widespread,

- resulting in substantial deviation at the genome wide scale. The enrichment for GxE at SNPs with high LDscore might be a hint on that hypothesis.
- c. Using the residual of the outcome adjusted for WHR (line 311-320) makes the quantification of potential bias a little more challenging, but in principle should result in similar effect. BTW, I might have missed it, but it wasn't clear to me whether the residual-based step was also used in the simulation or only in the real data analyses.
 - d. Altogether, I believe it would be difficult to fully exclude potential bias when using WHR as an environmental exposure. And based on the available data, I cannot say whether the reported real data results are valid, partly biased, or severely biased. Using a "standard" environmental exposure, i.e. with a likely causal effect on the outcome would be much more convincing.
3. The authors conducted a useful simulation study to validate the performances of MonsterLM. However, I found the scope of the simulation to be quite modest. As for any novel method, I would expect a broader range of scenarios to be investigated, including at least two additional analyses:
 - a. The simulation includes single block and 3-blocks analyses. They should also run a full genome (all 60 blocks used in the real data analysis). I appreciate the computational burden of such an analysis, but even a limited number of replicates (e.g. 20 instead of 100) would be informative on the variance of the estimate
 - b. As the real data application suggests that contribution of GxE varies with MAF and LD (supp Fig2), some simulations drawing effect conditional on those two parameters would help confirming the robustness of the approach conditional on these parameters.
 4. I am a puzzle with some of the R^2 estimates from figure 3. For example, for HDL, the R^2_E (~0.2), R^2_G (~0.25) and R^2_{GxE} (~0.6) sum up to 1+. Not sure if I am missing something, otherwise it would mean that the variance of HDL is basically fully explained by all three terms, which sounds unrealistic. Please clarify or revise. Full genome simulation under various scenarios (see comment 3) would be helpful in validating the analytical standard deviation of the approach. Note that based on the method description I do not see why there would be an issue there, and again I am mostly worried that the large R^2_{GxE} only reflects bias due the use of WHR as an exposure.
 5. The reported PS are also not much in agreement with the reported R^2_{GxE} . For example, it is challenging to explain how the GxE interaction on HDL explain >50% of the variance, but are providing only very modest improvement in prediction as compared to the marginal genetic effect, which itself explain "only" 25% of the variance. Again, a likely explanation is a potential bias due to the use of WHR.

Minor comments.

6. Triglycerides, CRP, and HDL are presented in the abstract and used in Figure 3, but then disappear from further figures. Some explanation or harmonization are welcome.
7. Line 136 "As expected, all heritability estimates were significant..." may be an overstatement?
8. Line 141-142: "*SNPs with ... and higher LDscore to disproportionately contribute to interaction variance*". The current consensus is that causal variants would tend to have low LDscore (PMID= 29700474). If not explain by potential bias (see comment 4b), some discussion about this is welcome.
9. A potentially interesting additional and simple metric to report to assess the link between marginal genetic effect and interaction (Fig 4 and supp Fig3) is the correlation –i.e. $\text{cov}(\beta_G, \beta_{GxE})$

10. Line 219-200: “existing methods are susceptible to bias and often yield vastly different estimates even when applied to the same data^{10–12}.” Looking at e.g. Fig1 from Ewans et al. (ref #12), I think that statement is a too strong. There are differences, but there are now much better understand and the various existing corrections produce fairly similar estimates.

11. I assume this is the case, but please confirm the R^2_G are derived using the residualized outcomes (i.e. adjusted for the exposure)

#####

Toy R simulation illustrating potential bias

N = 1000

Ns = 1000

b1=b2=b3=matrix(NA,Ns,2)

for(s in 1:Ns){

 G = scale(rbinom(N,2,0.3))

 U = scale(rnorm(N))

 # E1, an exposure varying with Y and harboring marginal genetic effect

 E1 = scale(U + rnorm(N)) - 0.1*G

 Y1 = scale(U + rnorm(N))

 # E2, as E1, but with heterogeneous genetic effect, e.g. GxSex interaction, but other models involving e.g. the tested SNP is only correlated with (one or multiple) causal variants would produce similar results

 S = scale(rbinom(N,1,0.3))

 E2 = scale(-U + rnorm(N)) - 0.5*G*S

 Y2 = scale(U + rnorm(N))

 # E3, a causal exposure to Y harboring various type of genetic effect

 E3 = scale(rnorm(N)) + 0.5*G - 0.5*G*S

 Y3 = scale(U + rnorm(N)) + E3

 b1[s,]=summary(lm(Y1~E1+G+G*E1))\$coef[3:4,1]

 b2[s,]=summary(lm(Y2~E2+G+G*E2))\$coef[3:4,1]

 b3[s,]=summary(lm(Y3~E3+G+G*E3))\$coef[3:4,1]

}

boxplot(cbind(b1,b2,b3), names=c("E1-G", "E1-GxE", "E2-G", "E2-GxE", "E3-G", "E3-GxE"))

abline(h=0, col="blue")

grid()

**RE: Article second submission (ID NCOMMS-21-15930). A versatile, fast and unbiased**
**method for estimation of gene-by-environment interaction effects on biobank-scale**
**datasets.**

**General Comments:**

Thank you for giving us the opportunity to submit a revised draft of the manuscript, “A versatile,
fast and unbiased method for estimation of gene-by-environment interaction effects on biobank-
scale datasets”, for consideration to be published in *Nature Communications*. We are grateful to
the reviewers for their time and effort in providing incredibly constructive suggestions and thank
them for their patience while we improved the manuscript. Prompted by the insightful feedback
from reviewers, we have extensively revised our manuscript and incorporated several major
analyses to significantly enhance the robustness of the MonsterLM methodology for estimation
on biobank scale GxE. Given the extent of modifications and the value added, we hope reviewers
are understanding of the time to revision. We have addressed the suggestions and comments
made by the reviewers in three sections: 1) a summary of the major amendments to the
MonsterLM method, 2) a response to shared concerns between multiple reviewers, and 3) a
point-by-point response to remaining individual reviewers’ comments and concerns. Reviewers’
comments are denoted with blue font, while the author responses are in black font.

**Section 1: Summary of the Major Amendments to the MonsterLM Method**

Attached is an adapted version **Figure 2** in the revised manuscript. This figure provides a step-
by-step illustration of how the revised MonsterLM method processes an environment exposure,
phenotype outcome, interaction, and genotype matrices. Methodological amendments since the
first submission are highlighted in pink:

QNTN: Quantile Normalize; STD: standardize to mean = 0 and SD = 1

 The method is described in detail in the main text’s Methodology section. Highlighted here are
 six additions to the method that are discussed throughout this response document. Briefly, (i)
 describes removing the need for imputation in the dataset, (ii) is a covariate adjustment with
 quantile normalization before and after regression, (iii) is a heteroskedasticity adjustment

function for the outcome regressand, (iv) enabling a CPU least squares estimator for users
without GPUs, (v) estimating additive genetic variance and GxE variance as separate matrices,
and (vi) providing a scaled adjustment factor for outcome regressands with non-null exposure
variance explained.

Furthermore, **Figure 2** of the main text discusses how to apply MonsterLM when using traits
with dichotomous exposures and/or outcomes.

**Section 2: General Responses to Shared Reviewer Comments**

In this section we respond to four themes raised by multiple reviewers: (1) type I error
susceptibility in GxE estimation, (2) model performance upon genome-wide simulation testing
across a more diverse range of scenarios, (3) benchmarking the heritability and GxE estimates
with other models in the literature, and (4) accounting for collider biases that could occur when
applying the method with a heritable exposure.

**Theme 1: Type I Error Susceptibility in GxE Estimation**

Rightfully, reviewers expressed some level of skepticism when the GxE estimates for some
blood biomarkers were greater than the additive genetic variance (or heritability) estimates.
Several hypotheses into potential biases were suggested thus prompting a thorough review.
Outlined below are two biases that were uncovered and subsequently corrected for in the revised
manuscript.

**Bias Source 1: Uncontrolled Heteroskedasticity**

When genome-wide simulations were conducted we noted inflated results compared to our set
point for GxE estimates but not for the additive genetic variance. This inflation was subtle for a
single block but became apparent when summed over 60 blocks in genome-wide analyses, as it it
amounted to up to a 5-fold increase in estimated GxE for a simulated set (i.e. $R_{GxE_set}^2 = 0.1$ and
$R_{GxE_estimate}^2 = 0.5$).

We narrowed down the source of this inflation to the effects of uncontrolled heteroskedasticity
occurring when fitting the interaction matrix to the phenotype outcome. Ordinary least squares
estimators assume constant variance of residuals, and violation of homoskedasticity can result in
inflated type I error. Further complicating matters, this issue appears unique to interactions and
not the additive genetic component, so heritability models do not conventionally adjust for this
explicitly.

To solve this, we applied an adjustment incorporating rank-level standardization on the
phenotype outcome sorted by strata of exposure levels. This procedure ensures the variance of
the outcome is minimally affected by the exposure. The inputs are the phenotype outcome and
exposure after the adjustments illustrated in **Figure 2** are applied. “Adjust Y for
heteroskedasticity” in this figure refers to applying the following function in R:

```
rm_Heteroscedasticity = function(P_resid, E_final)
{
nb.partitions <- 20
partition.size <- 1 / nb.partitions / 2
E_ranks <- rank(E_final, ties.method = "average") / (length(E_final) + 1)
P_resid_final <- P_resid
for( i in 1:nb.partitions) {
lower_lower_bound <- ( i - 1 ) * partition.size
lower_upper_bound <- i * partition.size
upper_lower_bound <- 1 - i * partition.size
upper_upper_bound <- 1 - ( i - 1 ) * partition.size
select <- which( (E_ranks >= lower_lower_bound & E_ranks < lower_upper_bound ) | (
E_ranks > upper_lower_bound & E_ranks <= upper_upper_bound ) )
P_resid_final[select] <- standardization( P_resid_final[select] )
}
return(P_resid_final)
}
```

We found “*nb.partitions* <- 20” to be the optimal number of partitions after simulation testing.

Step one in the main text Methods section further describes this adjustment along with the other
processing steps outlined in **Figure 2**.

Bias Source 2: Concurrent missingness of exposures and outcomes

In revisiting MonsterLM analyses, we discovered that correlated missingness between exposure
and outcomes also inflated type I error in the first submission for a subset of biomarker results
(Total bilirubin, Apolipoprotein B, and HDL-Cholesterol). If either exposure or phenotype
outcome data is missing and mean imputed, then estimations tend toward the null. However, if
missing data occur in the same individuals for both exposure and phenotype outcome, with
missing values mean-imputed then there can be severe type I error.

Accordingly, genome-wide simulations were designed for three imputation scenarios described
in lines 281 – 284, with corresponding results described in lines 400 – 406 of the manuscript and
in **Supplementary Table 5** (attached below).

Supplementary Table 5. MonsterLM Performance with Varying Imputation Conditions

Model (R^2_{set})	20% Y Imputation Average R^2 (σ)	20% E Imputation Average R^2 (σ)	20% E, Y Imputation Average R^2 (σ)	0% E, Y Imputation Average R^2 (σ)
G (0.2)	0.1660 (0.0046)	0.2071 (0.0047)	0.1660 (0.0046)	0.2054 (0.0047)
$G \times E$ (0.1)	0.0717 (0.0045)	0.0660 (0.0045)	0.8969 (0.0054)	0.0908 (0.0045)

**Imputation Scenarios.** MonsterLM performance using ten base model genome-wide (scenario 2)
simulations with specific imputation conditions. Average heritability estimates (G) compared to the
set point of 0.20 and $G \times E$ estimates compared to the set point of 0.10 under the following four
imputation conditions: i) 20% of the phenotype (Y) is mean imputed, ii) 20% of the exposure (E) is
mean imputed, iii) 20% of the same E and P individuals are mean imputed, and iv) no E or P mean
imputation.

The $G \times E$ estimation was inflated (marked in red) when mean imputation was used for
individuals with missing data for both the exposure and outcome in the same individuals. While
not as high as 20% of individuals, this data structure was observed in three inflated biomarkers
of the first submission.

To solve this, we remove individuals with missing exposure or phenotype outcomes, as noted in
the fourth line of **Figure 2**.

**Theme 2:** Genome-wide simulations across a wider range of scenarios

The robustness of MonsterLM was more comprehensively validated across 16 scenarios. This
presents 12 additional scenarios compared to the original submission's 3 scenarios. Furthermore,
each scenario is simulated 10 times genome-wide (1,030,579 SNPs and 325,989 individuals).
Briefly, scenario types include varying interaction effect distributions, effect SNP sources, effect
SNP distributions, binary exposures and outcomes, collider conditions, MAF/LD strata, and
imputation scenarios. They are described in detail in the methods lines 267 – 279, and the results
of the simulations display well-controlled bias and precision and are presented in lines 368 – 406
and in **Figure 3**.

In **Figure 3** additive genetic variance is set at 0.2 for every scenario. $G \times E$ variance is set at 0.1
for scenarios 2 – 10 the upper bounds of $G \times E$ for real data. $G \times E$ variance is set at 0 to test
calibration of null $G \times E$ variance for scenarios 1, 11, and 12.

Our main genome-wide simulations in **Figure 3** are shown below:

**Figure 3 | Estimation of variance explained by GxE for 12 simulated scenarios.** Estimation
 of variance explained by GxE for 12 genome-wide scenarios from 10 simulations. “None”
 indicates the absence of a condition. *Model*: with or without collider features. Dashed red lines
 indicate true set G (R^2_{GW}) variance and dashed blue lines indicate true set GxE variance
 (R^2_{GWEI}). β_{GE} Conditions: $LD > Q_3$: all β_{GE} effects were sampled from GxE effect SNPs in the
 highest LD quartile; $MAF < Q_1$: all β_{GE} effects were sampled from GxE effect SNPs in the
 lowest MAF quartile; sampling distribution for β_{GE} other than $\sim N(0,1)$ is denoted; $R^2_{E \text{ on } Y}$:
 outcome variance explained by exposure; E continuous unless otherwise stated; E Heritability:
 additive or heterogeneous. Scenario conditions toggle these parameters: (i) estimation in the null
 base scenario ($R^2_{GWEI} = 0$), (ii) estimation in the non-null base scenario ($R^2_{GWEI} = 0.1$), (iii)
 estimation when the exposure variance is raised to 0.1, (iv) estimation when β_{GE} is sampled from
 $LD \text{ SNPs} > Q_3$, (v) estimation when β_{GE} is sampled from $LD \text{ SNPs} > Q_3$ and $MAF \text{ SNPs} < Q_1$,
 (vi - vii) estimation when the assumptions of standardization for β_{GE} effects were invalidated by
 generating effects with exponential and beta distributions (positive kurtosis), (viii) estimation in
 scenario (i) but using a dichotomous generated E_{sim} , (ix) estimation in the collider scenario
 where β_{GE} and β_G effects were randomly selected, (x) estimation in the collider scenario where
 β_{GE} effects were not an element (completely non-overlapping) of β_G effects, (xi) estimation in
 the collider scenario where β_{GE} effects are a strict subset (completely overlapping) of β_G effects,
 and (xii) estimation in the collider scenario where simulated exposures are heritable through
 additive and heterogenous genetic effects. Means and 95% CIs are represented by dot and
 whisker plots per scenario. Each black dot represents a single simulation.

**Theme 3:** Benchmarking the novel MonsterLM method versus other heritability and GxE
 methods

 We additionally benchmarked MonsterLM heritability to alternative models (BOLT, GRE, mtg2)
 using the same input and outcome data to provide a fair comparison. mtg2 GxE was compared to
 MonsterLM for estimation of interaction effects. However, because of mtg2 computational
 constraints, sample size had to be reduced to 75,000 participants. Methodology for
 benchmarking is described in lines 356 – 358 and the results are presented in lines 454 – 473.
 Generally, MonsterLM heritability is comparable to other models and more conservative in some
 cases. We note similar WHR GxE estimates in traits such as cholesterol, height, and total
 bilirubin but traits such as glucose and HbA1c have higher estimations using mtg2 as shown in

**Table 3:**

Table 3. Benchmarking MonsterLM Additive Genetic Variance (Heritability, h_G^2) and GxE (GxE, $R^2_{GxE_{WHR}}$)

Trait	MonsterLM M h_G^2 (σ)	mtg2 h_G^2 (σ)	Bolt h_G^2 (σ)	GRE h_G^2 (σ)	MonsterLM $R^2_{GxE_{WHR}}$ (σ)	mtg2 $R^2_{GxE_{WHR}}$ (σ)
Apolipoprotein A	0.281 (0.0046)	0.220 (0.0061)	0.240 (0.0025)	0.295 (0.005)	0.0236 (0.0043)	0.00450 (0.0027)
Apolipoprotein B	0.219 (0.0045)	0.194 (0.0069)	0.232 (0.0023)	0.274 (0.005)	0.0242 (0.0043)	0.00410 (0.0030)
Cholesterol	0.179 (0.0045)	0.157 (0.0065)	0.160 (0.0022)	0.198 (0.005)	0.0180 (0.0042)	0.0104 (0.0032)
CRP	0.238 (0.0046)	0.0706 (0.0060)	0.228 (0.0023)	0.283 (0.005)	0.0711 (0.0044)	0.0005 (0.0031)
Glucose	0.105 (0.0047)	0.0447 (0.0053)	0.0903 (0.0019)	0.116 (0.0049)	0.00475 (0.0046)	0.2098 (0.0052)
HbA1c	0.289 (0.0047)	0.129 (0.006)	0.246 (0.0024)	0.284 (0.0051)	0.00884 (0.0044)	0.1392 (0.0045)
HDL-Cholesterol	0.313 (0.0044)	0.232 (0.0057)	0.279 (0.0024)	0.326 (0.0052)	0.0380 (0.0041)	0.0063 (0.0024)
LDL-Cholesterol	0.173 (0.0045)	0.153 (0.0066)	0.157 (0.0022)	0.195 (0.0050)	0.0194 (0.0043)	0.0072 (0.0031)
Triglycerides	0.205 (0.0042)	0.187 (0.0060)	0.229 (0.0023)	0.264 (0.0051)	0.0680 (0.0041)	0.0213 (0.0029)
Height	0.683 (0.0045)	0.316 (0.0062)	0.548 (0.0022)	0.497 (0.0054)	0.00136 (0.0039)	-0.0189 (0.0020)
Total Bilirubin	0.399 (0.0046)	0.431 (0.0077)	0.369 (0.0023)	1.346 (NA)	0.000634 (0.0043)	-0.0002 (0.0028)

**MonsterLM Benchmarking.** MonsterLM is compared to other heritability models. Presented
 are both additive genetic variance and GxE_{WHR} estimates with calculated standard deviations for the
 biomarkers and traits used in this study. The same genotypic and phenotypic individual-level data
 (N=325,989) is used except for mtg2 which was calculated on a smaller subset of N=75,000.

Theme 4: Addressing Effects of a Collider Bias on MonsterLM

We address effects of a collider bias under realistic but extreme conditions in the revised manuscript (lines 247 – 265):

“Secondly, we considered models including a collider bias. A collider bias can occur when two conditions are met: a controlled-for environmental exposure is both heritable and influenced through an unobserved confounder; and that same unobserved confounder influences the outcome (Figure 1; top panel). As shown in Aschard et al. 2015¹ and Akimova et al., 2021², a collider bias can potentially lead to over-estimation of GxE variance and other variance components. Collider model simulations had a set correlation R^2 between Y_{sim} and E_{sim} of 0.2 and assumed that the correlation was entirely due to the simulated unobserved confounder covariate (UC_{sim}), an extreme collider model (Figure 1; top panel). In these scenarios, E_{sim} was simulated to have 20% of its variance explained by additive genetic effects (i.e. heritability of E_{sim}), as WHR was observed to have similar heritability empirically. In one collider scenario, genetic heterogeneity³ (which has been explored in the context of GxE models), explained 20% of the genetic variance of E_{sim} (with the remaining genetic variance of E_{sim} explained by additive genetic effects). Genetic heterogeneity was simulated by including a genetic interaction with a randomly generated binary variable. Genome-wide simulated variances were set at $R^2_{GW} = 0.20$ (i.e. heritability of Y), and $R^2_{GWEI} = 0.1$ or 0.0 (i.e. interaction variance of Y). In base model simulations, observed correlations between E and Y are due to the true set value (i.e. causal effect of E on Y). In collider simulations, there is no causal effect of E on Y or Y on E such that any observed correlation is entirely due to the collider bias.”

Our simulated collider bias represents a rather extreme case where the simulated $E - Y$ correlation (0.45) is greater than the highest observed WHR-outcome correlation (Triglycerides; 0.33) and the simulated $E - Y$ correlation is entirely due to the collider bias (Figure 1 top panel; inset below).

In the above inset: green circles are outcome variables; red circles are predictor variables; red
 squares are colliders; blue circles are confounders; green arrows are causal associations; grey
 arrows are unobserved causal associations.

 We show that MonsterLM is robust and well-controlled in the face of a set of strongly simulated
 collider bias scenarios. Manuscript lines 384 – 386 discuss the simulation results modelling four
 variations of the collider bias (**Figure 3**; scenario 9 – 12; page 6):

 “Accurate GxE estimations were observed in the four scenarios including collider biases (**Figure**
 **3**; scenario 9-12). Accurate G estimations were observed in all collider scenarios except when
 heterogeneous exposure heritability was included (**Figure 3**; scenario 9 – 12).”

 As suggested by reviewers, to further confirm reported interactions are not the result of collider
 bias, we additionally tested a trait that, like WHR, reflects an obesogenic environment yet should
 be impervious to collider bias. The selected trait (M10; days of at least 10 minutes moderate
 exercise in a week) has negligible heritability ($R^2_{GW} = 0.02$) and is minimally correlated to tested
 phenotypes ($\{R^2_{E on Y} \in [0.00, 0.01]\}$), excluding collider bias as a source of GxE inflation.
 Results with GxE_{M10} showed significant interaction effects consistent with GxE_{WHR} as described
 in lines 421 – 424 and **Table 1** (shown below).

Table 1. Real Data Estimates for Continuous Outcomes with MonsterLM

Trait	Additive Genetic Variance (95% CI)	WHR GxE Variance (95% CI)	M10 GxE Variance (95% CI)	Smoking GxE Variance (95% CI)	Permuted Exposure GxE (95% CI)
-------	------------------------------------	---------------------------	---------------------------	-------------------------------	--------------------------------

Apolipoprotein A	0.281 (0.271 - 0.290)	0.0236 (0.014 - 0.033)	0.007 (-0.003 - 0.017)	0.000739 (-0.0193 - 0.0207)	-0.0027 (-0.0127 - 0.0073)
Apolipoprotein B	0.219 (0.210 - 0.228)	0.0242 (0.016 - 0.033)	0.040 (0.030 - 0.050)	0.00984 (-0.0102 - 0.0298)	-0.0038 (-0.0134 - 0.0062)
Total Cholesterol	0.179 (0.170 - 0.188)	0.0180 (0.010 - 0.027)	0.042 (0.032 - 0.052)	0.00505 (-0.0150 - 0.0251)	0.0000 (-0.0099 - 0.0099)
CRP	0.238 (0.229 - 0.247)	0.0711 (0.063 - 0.080)	0.028 (0.018 - 0.038)	0.0100 (-0.010 - 0.030)	-0.0022 (-0.0122 - 0.0078)
Glucose	0.105 (0.096 - 0.115)	0.00475 (-0.004 - 0.014)	0.003 (-0.007 - 0.013)	0.00776 (-0.0122 - 0.0278)	-0.0007 (-0.0106 - 0.0092)
HDL-Cholesterol	0.313 (0.303 - 0.323)	0.0380 (0.029 - 0.047)	0.010 (0.001 - 0.020)	-0.00234 (-0.0223 - 0.0177)	-0.0020 (-0.0120 - 0.0080)
HbA1c	0.289 (0.280 - 0.299)	0.00884 (0.000 - 0.018)	0.013 (0.003 - 0.023)	0.00610 (-0.0139 - 0.0261)	-0.0010 (-0.0110 - 0.0006)
Height	0.683 (0.674 - 0.694)	0.00136 (-0.007 - 0.010)	0.0047 (-0.005 - 0.015)	-0.00124 (-0.0212 - 0.0188)	-0.0013 (-0.0089 - 0.0063)
LDL-Cholesterol	0.173 (0.164 - 0.182)	0.0194 (0.011 - 0.0282)	0.045 (0.035 - 0.055)	0.00825 (-0.0118 - 0.0283)	0.0014 (-0.0085 - 0.0114)
Triglycerides	0.205 (0.196 - 0.213)	0.0680 (0.061 - 0.077)	0.024 (0.014 - 0.034)	0.00696 (-0.0130 - 0.0270)	-0.0006 (-0.0106 - 0.0094)
Total Bilirubin	0.399 (0.389 - 0.408)	0.000634 (-0.008 - 0.009)	0.002 (-0.007 - 0.012)	0.00498 (-0.0150 - 0.0250)	-0.0019 (-0.0119 - 0.0081)

**MonsterLM real data results.** Presented are real data estimates for continuous outcomes. All estimates
are performed using the MonsterLM methodology. Exposures include waist-hip-ratio (WHR), number of days
of 10 minutes moderate exercise (M10), dichotomous smoking status (Smoking), and permuted exposure
(E_{pm}). Bolded estimates indicate significant signal. Bolded estimates are significant.

This concludes a summary of the major revisions made for the resubmission to address shared
reviewer concerns. Henceforth, we address individual reviewer comments in a point-by-point
manner.

**Section 3: Individual Reviewers' Responses**

**Reviewer #1 (Remarks to the Author):**

A versatile, fast and unbiased method for estimation of gene-by-environment interaction
effects on biobank-scale datasets
Khan et al.

This paper proposes a novel GxE method that is an extension of GRE model (#19 in
references). It is basically a series of multiple regressions, each with a handful number of
SNPs (# SNPs < sample size) from which R^2 values are summed to estimate genome-wide
GxE variance. The authors applied the proposed method to the UK Biobank for 8 biomarkers
as the main phenotypes and waist-hip-ratio (WHR) as the environmental variable. The
phenotypic variance explained by estimated GxE ranged from 11 – 58%, which is

surprisingly large. The authors show a significant improvement of polygenic risk score
prediction when using their GxE model, compared to the null model (without GxE).

We thank the reviewer for this accurate assessment of our manuscript. We have since updated
these findings to include: (i) the reevaluation of 10 blood biomarkers and height where a more
modest 1 – 7% of GxE variance explained was estimated for significant biomarkers, (ii)
dichotomous outcomes (coronary artery disease and type 2 diabetes) and dichotomous exposures
(exercise, smoking, randomized exposure), and (iii) demonstration that only CRP polygenic
score prediction was improved with GxE.

This study is interesting and important as GxE is now widely recognised as an
important source of phenotypic variation. However, there are several comments and
questions that may help improve the current version of the manuscript.

1. The authors should verify their novel method using a wider range of simulation
scenarios. For example, the size of GxE variance (R^2) is 0.005 or 0.015 (Figure 2). Are the
estimates of the proposed model still unbiased when using a large GxE variance (e.g. 0.1 or
0.2)? Note that the authors reported that ~ 50% of phenotypic variance can be explained by
GxE.

We thank the reviewer for this important consideration concerning the robustness of GxE
variance estimation across a broader range of GxE scenarios. Section 2 Theme 2 displays the
revised genome-wide scenarios where many of them now use a large GxE variance (0.1).

Please also consider the genetic and residual correlations between the main phenotypes
and environmental variable. Ni et al (2019) reported that estimated GxE variance can be
biased when such correlations are not properly modelled.

We thank the reviewer for bringing up potential issues when genetic and residual
correlations between the main phenotypes and environmental variables are not properly
modelled. Please refer to Section 2 Theme 2 & 4 for discussion on this topic.

Briefly, genetic-outcome, genetic-environmental variable, and phenotype-environmental
correlations are modelled in the revised **Figure 3** base and collider scenarios. **Figure 2**
also shows the addition of an “adjustment for $\text{corr}(E,Y)$ ” which is now described in the
methods.

Ni et al. Genotype–covariate correlation and interaction disentangled by a whole-genome
multivariate reaction norm model. Nature communications 10, 2239 (2019)

We thank the reviewer for providing this excellent resource that we have added as a reference to
our manuscript (#50). Ni et al., 2019 reports “*We also show that the residual variances estimated*
*by standard additive models can be inflated in the presence of G–C and/or R–C interactions*”
highlighting the importance of the collider bias and challenges with adapting the additive model
to the field of interactions. We have added this reference when comparing to other GxE models
in **Table 4**.

The authors should also assess if the type I error rate is controlled under the null
model (in the absence of GxE).

We thank the reviewer for suggesting this informative experiment. The manuscript now includes
exclusively genome-wide simulations. Scenario 1, 11, and 12 in **Figure 3**, as described in
Section 2 Theme 2, assess MonsterLM type I error under the null model of no interaction effect.

2. Can this proposed method be applied to binary disease traits? If so, simulation of binary
variables should be used to verify the method. Otherwise, the authors should add a
limitation to clarify this. Please also note that there are a number of existing methods
(including GxEsum that can be based on GWAS summary stats) can be applied to disease
traits, which should be acknowledged in Discussion.

GxEsum: genotype-by-environment interaction model based on summary statistics. bioRxiv
(2020) doi: <https://doi.org/10.1101/2020.05.31.122549>

We thank the reviewer for this insightful inquiry. Indeed, we have now extended MonsterLM to
binary disease trait outcomes. We do this by applying some key changes to the MonsterLM
algorithm from Section 1 as shown by the changes highlighted in pink in the adapted **Figure 2**
below:

QN: Quantile Normalize; STD: standardize to mean = 0 and SD = 1

 **Figure 2** (adapted) | The MonsterLM method split into three stages for binary outcomes
 and continuous exposures. The first stage describes data processing, the second stage

describes methods of computing least squares, and the third stage describes how to finalize
 estimates and compute confidence intervals.

 Modifications to the algorithm labelled in pink font above include (i) the use of “Y stratified QN
 E” which means to quantile normalize the continuous exposure in each binary outcome group
 separately (Step 1; E), (ii) standardizing ($\mu = 0$, $\sigma = 1$) the binary outcomes (Step 1; Y), and (iii)
 and applying a liability scale transformation on the per block estimates (Step 2).

 The methodology is discussed in the manuscript lines 228 – 232, simulation testing methodology
 in lines 286 – 288, results for binary disease outcome simulations in lines 396 – 398, and results
 for a real data application on Type 2 Diabetes and Coronary Artery Disease in 428 – 436.

 We thank the reviewer for directing us to other GxE models that can handle binary data. We
 have added the GxESum method and the MTG2 IGE method to **Table 4** (subset shown below)
 comparing MonsterLM to current GxE estimation methods:

Table 4 Subset. Comparison of current methods estimating gene-by-environment contributions to MonsterLM

Method	Description	Advantages	Limitations	MonsterLM
GxESum ⁴	Estimates genome-wide GxE variance using GWAS summary statistics	Has controlled type I error rates and unbiased GxE estimates, efficient computation-wise. Can also be applied to binary disease traits.	Disadvantages of using summary statistics include information limitations and population stratification bias susceptibility.	Advantages of individual-level data include better handling of LD and complex interactions.
MTG2 IGE ^{5,6}	Estimates variances explained by additive effects of exposure variables, by ExE interactions, and covariance between genetic effects and exposomic effects	Precise estimates with low standard deviations, flexible for a variety of exposure-based interactions.	Requires generation of a genomic relationship matrix file (grm file) using plink. This matrix becomes substantially large when the number of participants reaches more than 100K. The file becomes the size of terabytes.	Does not require generating a genomic relationship matrix, and is well suited for large numbers of participants in biobank-scale data

**Existing methods of biobank GxE estimation compared to MonsterLM.** A description of six
 methods to estimate GxE estimates with respective advantages, limitations, and comparison to
 MonsterLM.

3. Can this proposed method be applied to binary or ordinal categorical environmental
 variables? Although the authors used WHR (continuous) as the environmental variable, a
 substantial proportion of environmental variables is not continuous. The authors should
 comment or discuss on this issue with proper references (several GxE models may allow
 using binary or ordinal categorical environmental variables).

We thank the reviewer for this inquiry. Indeed, MonsterLM has since been developed to be
applied to binary exposures as well. An adapted **Figure 2** outlines our binary exposures approach
with key changes highlighted in pink:

QN: Quantile Normalize; STD: standardize to mean = 0 and SD = 1

**Figure 2** (adapted) | The MonsterLM method split into three stages for continuous outcomes and
binary exposures. The first stage describes data processing, the second stage describes methods
of computing least squares, and the third stage describes how to finalize estimates and compute
confidence intervals.

Modifications to the algorithm include (i) no exposure (E) adjustment (Step 1; E), (ii) the use of
“ E stratified QN Y ” which means to quantile normalize the continuous phenotype in each binary
group separately (Step 1; Y), and (iii) standardization ($\mu = 0$, $\sigma = 1$) of the interaction (GxE)
terms (Step 1; GxE).

The methodology is discussed in the manuscript lines 222 – 226, simulation testing methodology
in lines 274 – 275, results for binary exposure simulations in lines 396 – 397, and results for a
real data application with smoking history as a binary exposure in lines 422.

4. In the simulation, the authors used a single chromosome to verify their method. The
authors should use all chromosomes because the proposed model may not be able to
account for correlation between chromosomes if there is any. This is also a concern for the
original GRE paper (that seems using a single chromosome to verify GRE model). Even with
samples after relatedness cut-off QC, correlation among chromosomes can be non-
negligible (see Figure 1 in the following reference).

Estimation of genomic prediction accuracy from reference populations with varying
degrees of relationship (2017) PLoS ONE 12: e0189775.

The reviewer raises an excellent consideration for the importance of genome-wide
simulation testing. Accordingly, we have investigated this thoroughly as detailed in
Section 2 Theme 2 where simulated outcomes are generated using genetic data across all
autosomes. Furthermore, we added this reference (#24) to further justify the importance of
choosing an unrelated cohort for our study design in lines 130 – 133 of the manuscript:

“*We selected 325,989 unrelated British individuals from the UK Biobank with both genotype and*
*biomarker data for inclusion in the analysis. A strictly unrelated set of individuals were chosen*
*to reduce genomic prediction inaccuracies²⁴.*”

My point is that the sum of estimates across chromosomes can be overestimated unless
correlations among chromosomes are properly modelled. The authors should revisit their
estimates from the real data (up to 58% of phenotypic variance).

We appreciate the reviewer’s hypothesis that chromosomal relatedness across a population level
could be a contributing source of bias that could lead to overestimation, as our reported GxE was
as high as 58% in the first draft. We now report GxE variance estimates ranging from 0 to 7%, as

mentioned in the first comment response. We discovered the two bias sources described in
Section 2 Theme 1 as being the source of the high GxE estimates.

5. Line 214-220. Did the author really investigate their estimates when varying parameters
that determines the genetic architecture? For example, the authors should check if the
estimates are robust to different values of scale factors used in reference #9, 10 and 11
(specifically alpha in equation (1) in reference #10). Please also see supplementary Table
12 in Ni et al Nature communications 10, 2239 (2019) that shows estimated GxE variance is
robust to misspecified scale factors.

We thank the reviewer for raising this consideration. Simulations in our initial submission
assumed effect sizes follow standard normal distributions for genetic, environmental and GxE
effects. As Speed et al., 2017 and Ni et al., 2019 point out, models cannot be assumed to be
robust in the face of varying MAF and LD genetic architecture. We added genome-wide
simulations scenarios where the distribution of interaction SNP effects follow non-normal
distributions. MonsterLM GxE is robust to the violation of scale factor assumptions since:

i) Genome-wide simulations of scenario 6 and 7 are accurate where β_{GE} effect distributions
are generated as exponential ($\beta_{GE} \sim exp(2)$) and with a positive kurtosis beta distribution
($\beta_{GE} \sim Beta(1.5,10)$), respectively. Note that scenarios 6 and 7 model effects as non-
Gaussian distributions. MonsterLM GxE estimations are accurate when violating these
scale factor assumptions as depicted in the results of the manuscript lines 381 - 383
(**Figure 3**; page 6).

ii) Speed et al., 2017⁷ comment on the robustness of models across varying MAF and LD
structures and that changing assumptions of SNP effects with Gaussian distributions can
lead to variability in estimate control depending on the MAF and LD architecture. Ni et al.,
2019³ supplementary table 12 shows “*Estimated genetic variance and SNP-heritability
may be biased if an assumption of the variance due to causal variants across different
MAF spectrums is violated*”. To assess how MAF and LD architecture could affect
MonsterLM estimates, we conducted the real-world data GxE_{WHR} analysis within 20 MAF
and LD bins for the 8 biomarkers with significant GxE effects (manuscript lines 447 – 451;
**Supplementary Figure 3**; shown below):

**Supplementary Figure 3** | Results for WHR are stratified by MAF and LD conditions. Each
significant interaction biomarker has a per SNP estimate for one of twenty stratified MAF and
LD conditions. The top panels (G) per biomarker is the heritability stratified estimate per SNP
and the bottom panels (GxWHR) per biomarker is the $G \times E_{WHR}$ stratified estimate per SNP. SNP-
adjusted R^2 is the adjusted R^2 divided by the number of SNPs per stratum.

We further assessed MonsterLM performance across a range of MAF and LD bins by
performing genome-wide scenario 2 (**Figure 3** legend) simulations (manuscript lines 391 – 394;
**Supplementary Table 3**). This is discussed in the 3rd comment of reviewer #3. We do not
observe any contribution bias relative to other groups at any of the MAF/LD levels for both real-
world data and simulation testing.

6. In relation to the question above, the authors mentioned the proposed method makes no
assumption with respect to the genetic (heritability) model (line 227). However, it uses
standardised genotype coefficients, i.e. scale factor (alpha) = -0.5 (please see equation (1)
in reference #10). Would the proposed method be able to get the same results (in terms of
R^2) without standardisation?

We thank the reviewer for pointing this out. We do make this assumption to our genetic
(additive) heritability model. However, we do not make further assumptions on our LD and
MAF predictors. We have revised lines 565 – 566 of the manuscript to acknowledge this:

“*MonsterLM assumes an additive genetic model and does not apply further parametrization for*
*underlying assumptions.*”

We refer to part (i) of the previous response to illustrate that MonsterLM is still robust when the
genetic scale factor is changed to a non-normal distributions. We also note that our additive
genetic model provides comparative heritability estimates to BOLT, GRE, and mtg2 (**Table 3**).

<Minor comments>

I found a number of references are not very relevant, e.g. in line 137, Please check if the
reference #16 reported any estimated SNP-heritability for those traits. In line 224, is
reference #16 appropriate? The authors should carefully check references and their
appropriateness through the text.

Thank you for this suggestion. We have performed a thorough curation of the references and
have revised them as appropriate. Regarding this statement it has been revised in the manuscript
lines 432 – 433: “*Outcome heritability estimates for all 13 traits were significant and largely*
*consistent with published estimates and other methods (BOLT, mtg2, and GRE; Tables 2 & 3).*”

Line 222. Did the studies reported that their GxE estimates were biased? Or, did the authors
check and confirm that GxE estimates across those methods were significantly different
from each other, using the same data? Please provide more explicit evidence to support the
argument.

We thank the reviewer for this recommendation. We have amended our discussion to specifically
refer to biases across heritability estimates, as these studies examine different exposures used in
conjunction with traits like BMI (resulting in different estimates of GxE variance). We believe
these biases can also be applied to GxE variance, as both state-of-the-art methods require
parametrization of the genetic architecture. We amended the main text discussion lines 553 –
558:

“*In many settings, inference methods for genome-wide SNP-heritability and GxE make*
*assumptions about genetic architecture. These assumptions are parametrized by polygenicity*
*(the number of variants with effects) and MAF/LD-dependence (the coupling of effects with*
*MAF, LD or other functional annotations). Since the true genetic architecture of any given trait*
*is unknown, existing heritability methods may yield vastly different estimates even when applied*
*to the same data*⁷⁻⁹.”

In fact, this could be why GRE overestimates Total Bilirubin in **Table 3** as this trait has quite a
different genetic architecture compared to others.

Line 265. ‘... specific GxE interactions’ should be ‘GxE interaction for specific variants’?

Thank you. Revised as suggested by the reviewer.

What is the reason to select 8 biomarkers as the main phenotypes? Why not for height or
BMI that are well studied-model traits?

We thank the reviewer for this inquiry into the study design. The original study aimed to choose
some of the most clinically utilized blood biomarkers as a starting point to test real-world data.

We have since added apolipoprotein A, glucose, coronary artery disease, height, and type 2
diabetes to the main phenotypes.

Equation (3). I don’t think the vertical bar is a correct notation for this expression. Please
revise.

We thank the reviewer for considering the potential incorrect notation for this expression.

We have since corrected equation (3) in the manuscript as:

$$"U = G, \text{ or } U = GE" \quad (1)$$

as outlined in **Figure 2**.

Reviewer #2 (Remarks to the Author):

The basic idea of the manuscript is to apply the GRE method of Hou et al. 2019 to gene-
environment interactions. While I think this is potentially a good idea, the authors have not
convinced me that their method works as advertised.

1. A major concern I have is that they first residualise the phenotype for the
environmental exposure before fitting the gene-environment interaction models. When the
phenotype in question is highly correlated with the environmental exposure, as is likely to be
the case for much of their empirical analyses, this means that the variance components
they estimate are not representative of the variance components of the original phenotype
before residualisation.

It appears that fitting the environmental exposure jointly with the GxE interaction model
would only add 1 parameter to the model, so why not do this?

We thank the reviewer for this insightful observation. Indeed, the environmental exposure is just
 one parameter to the model and would not be computationally taxing to include per block
 estimate. However, there are two main advantages to residualizing the exposure, as is detailed
 below. We revised the MonsterLM algorithm to adjust for any biases created when the exposure
 is significantly associated with the outcome (i.e. $R^2_{E,Y} > 0$) while still preserving the advantages
 of residualizing the exposure. As the reviewer notes, the variance of the outcome after
 residualizing is not necessarily the same as before, so accounting for this is warranted.

We residualize the exposure for two reasons:

i) If we include the exposure as an additional parameter per block, we are effectively
 including the variance of $R^2_{E,Y}$ in each block estimation (**Figure 2**; Step 2) and summing
 across all blocks genome-wide, which can lead to an overestimation of the additive or
 interaction variance. Residualization accounts for the variance of $R^2_{E,Y}$ a single time per
 outcome.

ii) Likewise, if we include the exposure as an additional parameter in each block, we are
 including the potentially overestimated $R^2_{G_i}$ or $R^2_{GE_i}$ (where i is a block) in each
 variance of the squared multiple correlation coefficient estimate, as seen in equation (9):

$$561 \quad \widehat{Var}_\infty(R^2_{U_i}) = (1 - R^2_{E,Y}) \left(\frac{n-1}{n-m-1}\right)^2 \frac{(n-m-1)(n-m+1)}{(n^2-1)} \text{HyperG}(R^2_{U_i}, n, m) \quad (8)$$

which could overestimate the imprecision.

We have since applied an adjustment to correct for non-null $R^2_{E,Y}$ and verified it with genome-
 wide simulations and real-world data estimations. The correction is a scale factor multiplied by
 the total summed R^2 estimate and total imprecision estimate. This correction is seen in equations
 (7) – (9) as the term: $(1 - R^2_{E,Y})$. Below is the adjustment applied to total additive genetic
 (R^2_{GW}) and GxE interaction variance (R^2_{GWEI}), respectively (for j block estimates):

$$R^2_{GW} = (1 - R^2_{E,Y}) \sum_{i=1}^j R^2_{G_i} \quad (6)$$

$$R^2_{GWEI} = (1 - R^2_{E,Y}) \sum_{i=1}^j R^2_{GE_i} \quad (7)$$

From a practical standpoint, residualization with the adjustment term also makes matrix
 inversion simpler as there is no need to create a unique augmented (genotype or interaction block

plus exposure) matrix particular to a specific exposure. In other words, the same inverted matrix
can be applied to multiple exposures.

2. (a) I found the simulations performed to be inadequate. The simulations only consider a
single chromosome. It would be better to simulate a model based on genome-wide
SNPs, and to see how the method performs when summing over chromosomes. (b) It is
also unclear exactly how the authors' method of calculating adjusted R^2 compares to
the methods of Hou et al. 2019. Could the authors compare their heritability estimates to
those from the method of Hou et al.? (c) Further, Hou et al. split the genome by
chromosome, which should give approximately independent blocks in a homogeneous
population. Why don't the authors also split by chromosome? (d) Also, can the authors
demonstrate that their standard errors and confidence intervals are accurate?

We thank the reviewer for their comments and suggestions. We have labelled this comment with
sub headers (a) - (d).

(a) We have since amended the manuscript to only include genome-wide simulations, as
discussed in Section 2 Theme 2.

(b) We have since provided detailed benchmarking to our manuscript. **Table 3** of the main text
and Section 2 Theme 3 of this document describe the benchmarking analyses.

(c) We thank the reviewer for commenting on the method of SNP partitioning. MonsterLM aims
to utilize as many SNPs per chromosome as possible to create independent blocks as the
reviewer suggests. However, large chromosomes (e.g. 1 and 2) have just over 80,000 SNPs
(when using our MAF and LD filters) making computation of ordinary least squares
estimators challenging with contemporary computers (**Figure 2**; Step 2). We thus create
blocks of up to 25,000 SNPs to ensure adequate computational speed, as noted in
**Supplementary Table 1**. However, we do see full chromosome computation as a possibility
in the foreseeable future as hardware technology continually improves and study sizes are also
getting larger.

MonsterLM computed the genome-wide estimates using 60 blocks to minimize
intrachromosomal junctions. Small chromosomes, such as chromosome 22, have no such
junctions and the largest chromosomes have a maximum of three intrachromosomal junctions
(4 blocks on the chromosome). The total number of intrachromosomal junctions is 38 (with
20 of them being centromeric locations) and we find it to have negligible impact on additive
genetic variance estimation and interactions when accounting for simulations (**Figure 3**) and
model benchmarking (**Table 3**).

We have since made two amendments to the manuscript to make the SNP partitioning
 methods clearer:

- i) Lines 138 – 140 of the manuscript now states: “Genetic variants were partitioned to
 minimize the number of blocks on each chromosome, with each block having a
 maximum SNP count of 25,000.”
- ii) The top panel of **Figure 1** includes an infographic showing SNP partitioning for
 chromosomes 1 and 22, as seen in the inset below:

- (d) We thank the reviewer for the suggestion of including an additional analysis to quantify the
 precision of MonsterLM. We have since added a results section to the manuscript where
 precision estimates are discussed (lines 374 – 376). We demonstrate the accuracy of
 confidence interval calculations by comparing observed variances in our simulated genome-
 wide estimates to the predicted variances described in our formulae (**Supplementary Table**
 **2** shown below). Furthermore, they can be compared to other models in **Table 3**.

Supplementary Table 2. Predicted Versus Observed Variance Estimates to Compute MonsterLM Confidence Intervals

Scenario	Predicted Variances Average ($\bar{\sigma}_{G_{pred.}}^2$) and Observed Variances ($\sigma_{G_{obs.}}^2$) for	
	Heritability (G) and Interaction ($G \times E$) Simulation Estimates	
	G : ($\bar{\sigma}_{G_{pred.}}^2, \sigma_{G_{obs.}}^2$)	$G \times E$: ($\bar{\sigma}_{G \times E_{pred.}}^2, \sigma_{G \times E_{obs.}}^2$)
$2.12 \times 10^{-5}, 2.81 \times 10^{-5}$	$1.94 \times 10^{-5}, 1.04 \times 10^{-5}$
$2.16 \times 10^{-5}, 1.65 \times 10^{-5}$	$2.16 \times 10^{-5}, 1.52 \times 10^{-5}$
$1.97 \times 10^{-5}, 2.81 \times 10^{-5}$	$1.84 \times 10^{-5}, 2.22 \times 10^{-5}$
$2.18 \times 10^{-5}, 3.06 \times 10^{-5}$	$2.04 \times 10^{-5}, 2.78 \times 10^{-5}$
$2.12 \times 10^{-5}, 1.39 \times 10^{-5}$	$2.04 \times 10^{-5}, 3.51 \times 10^{-5}$
$2.15 \times 10^{-5}, 4.40 \times 10^{-5}$	$2.04 \times 10^{-5}, 1.38 \times 10^{-5}$
$2.16 \times 10^{-5}, 3.22 \times 10^{-5}$	$2.05 \times 10^{-5}, 1.00 \times 10^{-5}$
$2.08 \times 10^{-5}, 2.65 \times 10^{-5}$	$2.06 \times 10^{-5}, 3.63 \times 10^{-5}$
$2.05 \times 10^{-5}, 2.03 \times 10^{-5}$	$3.19 \times 10^{-5}, 1.16 \times 10^{-5}$
$2.00 \times 10^{-5}, 1.78 \times 10^{-5}$	$3.22 \times 10^{-5}, 1.81 \times 10^{-5}$

$2.04 \times 10^{-5}, 2.84 \times 10^{-5}$	$3.21 \times 10^{-5}, 2.84 \times 10^{-5}$
$2.34 \times 10^{-5}, 1.19 \times 10^{-5}$	$2.03 \times 10^{-5}, 1.04 \times 10^{-5}$
Statistical Significance	$p = 0.188$	$p = 0.329$

**Precision Calibration.** MonsterLM precision concordance between the average predicted
variances of the simulated scenarios compared to their observed variance as per the MonsterLM
method. Predicted variances averages ($\bar{\sigma}_{pred.}^2$) and observed variances ($\sigma_{obs.}^2$) for heritability (G)
and interaction (GxE) simulation estimates are compared. Two sample t-tests are used to assess for
significant differences between the groups.

3. (a) The authors find very large interaction variance components. If true, this is an
important result. However, they are larger than the heritable components for some
traits, which I find somewhat implausible -- I wonder if this could be an artefact of first
residualising the biomarkers for waist-hip-ratio. (b) The authors show that most of their
signal is coming from SNPs without significant marginal effects. Do these SNPs also
exhibit effects on the phenotypic variance? As would be expected if they are involved in
interactions.

We thank the reviewer for their comments and suggestions.

(a) We agree with the reviewer's position that our GxE results in the original submission were
rather large and the magnitudes greater than what other GxE models report. We address this
in detail in Section 1 and Section 2 Theme 1 of this document. Regarding exposure
residualization, Reviewer 2 comment 1 explains how we corrected for residualization in
MonsterLM, and that the correction is not a source of Type I error.

(b) Upon revision of the MonsterLM method, secondary analyses were updated (**Figure 5;**
**Supplementary Figure 4 and 5**) and results presented in the manuscript (lines 477 – 514).
The results are discussed in manuscript lines 583 – 589:

*“Our results also provide some further insights into why identification of GxE has been*
*challenging. Many prior studies have reasonably focused the search for significant GxE*
*interactions on variants with genome-wide significant marginal effects. Our results show*
*that a majority of GxE effects are due to variants with unremarkable marginal effects (i.e.*
*only 3 – 28% of GxE recovered with SNPs with strong marginal effects), although variants*
*with strong marginal effects remain preferred candidates for GxE interactions. We also*
*show in a proof-of-concept experiment that incorporation of GxE can improve PS*
*prediction, albeit very modestly.”*

**Supplementary Figure 4** goes into detail on how much of each significant marginal and
interaction SNPs contributes to phenotypic variance recovery. **Supplementary Figure 6** and
**Supplementary Table 9** show that incorporation of SNPs with largest interaction effects
increases polygenic score prediction, albeit modestly, in proportion to the size of the overall
GxE estimate from **Table 1**. For instance, CRP and TG had the largest GxE_{WHR} estimates
and also show the greatest relative increase in polygenic scores predictiveness.

4. Could the authors compare their GxE heritability estimates to those from the method of
Ni et al. 2019 (Genotype–covariate correlation and interaction disentangled by a whole-
genome multivariate reaction norm model)?

We thank the review for this suggestion. Indeed, Ni et al., 2019 study provides a comparable method
to explore GxE estimates. One caveat to our direct comparison is that we found the method from Ni
et al., 2019 required a genome relationship matrix file that was too large (> 1TB RAM space) to
compute for our sample size of 325,989 individuals. We tried to maximize the number of individuals
in the mtg2 analysis and hence included 75,000 individuals. Benchmarking is described in lines 356
692 – 358 and lines 454 – 473. See **Table 3** (subset) below for the comparison of MonsterLM GxE
estimates to mtg2:

Table 3. Benchmarking MonsterLM Additive Genetic Variance (Heritability, h_G^2) and GxE (GxE, R^2_{GxE})

Trait	MonsterLM h_G^2 (σ)	mtg2 h_G^2 (σ)	MonsterLM R^2_{GxE} (σ)	mtg2 R^2_{GxE} (σ)
Apolipoprotein A	0.281 (0.0046)	0.220 (0.0061)	0.0236 (0.0043)	0.00450 (0.0027)
Apolipoprotein B	0.219 (0.0045)	0.194 (0.0069)	0.0242 (0.0043)	0.00410 (0.0030)
Cholesterol	0.179 (0.0045)	0.157 (0.0065)	0.0180 (0.0042)	0.0104 (0.0032)
CRP	0.238 (0.0046)	0.0706 (0.0060)	0.0711 (0.0044)	0.0005 (0.0031)
Glucose	0.105 (0.0047)	0.0447 (0.0053)	0.00475 (0.0046)	0.2098 (0.0052)
HbA1c	0.289 (0.0047)	0.129 (0.006)	0.00884 (0.0044)	0.1392 (0.0045)
High Density Lipoprotein	0.313 (0.0044)	0.232 (0.0057)	0.0380 (0.0041)	0.0063 (0.0024)
Low Density Lipoprotein	0.173 (0.0045)	0.153 (0.0066)	0.0194 (0.0043)	0.0072 (0.0031)
Triglycerides	0.205 (0.0042)	0.187 (0.0060)	0.0680 (0.0041)	0.0213 (0.0029)
Height	0.683 (0.0045)	0.316 (0.0062)	0.00136 (0.0039)	-0.0189 (0.0020)
Total Bilirubin	0.399 (0.0046)	0.431 (0.0077)	0.000634 (0.0043)	-0.0002 (0.0028)

**MonsterLM Benchmarking.** MonsterLM is compared to other heritability models. Presented are
both additive genetic variance and GxE estimates with calculated standard deviations for the biomarkers
and traits used in this study. The same genotypic and phenotypic individual-level data (N=325,989) is
used except for mtg2 which was calculated on a smaller subset of N=75,000.

We comment on the findings on lines 460 – 466 of the manuscript:

“*MonsterLM GxE estimates with WHR were compared to mtg2 (Table 3). The mtg2 analysis was*
*limited to 75,000 individuals due to computational constraints. GxE estimates were consistent*”

*between MonsterLM and mtg2 for cholesterol, height, and total bilirubin. However, glucose and*
*HbA1c had considerably higher GxE estimates with mtg2 compared to MonsterLM (0.210 and*
*0.139 in mtg2, versus 0.00475 and 0.00884 in MonsterLM). MonsterLM heritability estimates of*
*glucose and HbA1c were more consistent with Bolt and GRE versus mtg2 (0.105 and 0.289 in*
*MonsterLM, versus 0.045 and 0.129 in mtg2).”*

Heritability estimates between the two models differ for CRP, Glucose, HbA1c, and height, with
MonsterLM estimates being more concordant with BOLT and GRE. Again, it is unclear how much
of this is due to the sample size difference.

5. Finally, I worry that some of what they are observing is the result of a non-linear
relationship between phenotype and environmental exposure. If these phenotypes have
a non-linear relationship with WHR, and WHR is heritable, then we might expect to find
what appears to be GxE interactions but is in fact GxG interactions due to a non-linear
relationship between WHR and phenotype.

We thank the reviewer for this comment. Accounting for non-linear associations between the
phenotypes and WHR is a valid concern. We added two sets of analyses to the manuscript to
address this point:

i) Nonlinear association testing of WHR with each phenotype was performed. To assess
nonlinearity, we use a method that applies adaptive local linear correlation computation,
with non-linearity estimated as a nonlinear correlation estimate (between 0 and 1). The
method is cited here: Chitta Ranjan and Vahab Najari. “Package ‘nlcor’: Compute
Nonlinear Correlations”. In: Research Gate(2020).doi:10.13140/RG.2.2.33716.68480. The

table below shows the nonlinearity results. There was no significant nonlinearity between
 outcome-exposure combinations from **Table 1**. The table below displays these results:

Table. MonsterLM Exposure Variance ($R_{WHR,P}^2$) and Nonlinear Correlation Coefficients ($NCOR_{E,P}(P)$)

Trait	MonsterLM $R_{WHR,P}^2 (\sigma)$	MonsterLM $NCOR_{WHR,P}(P)$	MonsterLM $NCOR_{E_{pm},P}(P)$
Apolipoprotein A	0.0544 (0.00081)	0.000443 (0.81)	0.00236 (0.2)
Apolipoprotein B	0.0118 (0.00038)	0.000628 (0.72)	0.000925 (0.60)
Cholesterol	3.03×10^{-6} (4.28×10^{-6})	0.000128 (0.94)	0.000868 (0.62)
CRP	0.104 (0.00102)	0.00193 (0.27)	0.00208 (0.24)
Glucose	0.0171 (0.00047)	0.000820 (0.65)	0.00233 (0.20)
HbA1c	0.0467 (0.000738)	0.00229 (0.20)	0.000566 (0.75)
High Density Lipoprotein	0.108 (0.00107)	0.000533 (0.77)	0.000811 (0.66)
Low Density Lipoprotein	0.00234 (0.00017)	0.000406 (0.82)	0.00185 (0.29)
Triglycerides	0.131 (0.00111)	2.69×10^{-6} (0.99)	0.00442 (0.010)
Height	0.00302 (0.00019)	2.02×10^{-5} (0.98)	0.000224 (0.90)
Total Bilirubin	0.0128 (0.00039)	0.000129 (0.94)	0.00291 (0.10)

MonsterLM Nonlinear Correlation Testing. Coefficient of determination of the WHR-phenotype association using linear models in the UK Biobank The nonlinear correlation estimate was calculated using “nlcor” R package with an associated p-value ($NCOR_{WHR,P}(P)$). The correlation estimate is between 0 and 1, with higher values representing a more nonlinear correlation. P: phenotype; WHR: waist-hip-ratio;

ii) The reviewer also notes the nonlinear GxE effects could be a concern if the exposure is
 heritable. Our analysis with a minimally heritable obesogenic exposure (“M10”) is
 discussed in Section 2 Theme 4 of this document. We note consistent GxE estimates
 using this non-heritable exposure.

 **Minor comments:**

 Line 117: it would help to explain briefly the simulation scenarios in the main text

Thank you. Section 2 Theme 2 describes the increased detail in explaining each simulation type
in the main text through improved writing and visualization. Base and collider direct acyclic
graphs are now included in **Figure 1** and an extensive legend is applied in **Figure 3** to aid in the
communication of simulation scenario tests.

Line 285: were any further sample quality control filters applied beyond ancestry?

We have provided more details on the sample quality control filters applied below:

UKBiobank samples were genotyped on either the UK Biobank Array (~450,000) or the UK
BiLEVE array (~50,000). Further imputation was conducted by the UKBiobank study team
using a combined reference panel of the UK10K and Haplotype Reference Consortium datasets.
Imputed genotypes (version 3) for 488,264 UKBiobank participants were downloaded through
the European Genome Archive (Category 100319). Samples were removed if they were flagged
for any of the UKBiobank-provided quality control annotations (Resource 531;
“ukb_sqc_v2.txt”) for high ancestry-specific heterozygosity, high missingness, mismatching
genetic ancestry, or sex chromosome aneuploidy (“het.missing.outliers”,
“in.white.British.ancestry.subset”, “putative.sex.chromosome.aneuploidy”). Samples were also
removed if their submitted gender did not match their genetic sex or if they had withdrawn
consent at the time of analysis. Variant quality control consisted of removing variants that had
low imputation quality (INFO score ≤ 0.30), were rare (MAF ≤ 0.005), or were in Hardy
Weinberg Disequilibrium (HWE $P \leq 1 \times 10^{-10}$). The HWE test was conducted within a subset of
unrelated individuals for each ethnic strata.

Line 392: 'tested whether direction' -> 'tested whether the direction'

Thank you. This has been revised as suggested.

Methods section: The authors appear to have the effect vector left-multiplying the genotype
matrix. Shouldn't it be right-multiplying?

Thank you. This has been revised as suggested.

**Reviewer #3 (Remarks to the Author):**

The authors propose a new approach to estimate the contribution of GxE to variance of
quantitative outcomes in large dataset. The manuscript is well written and the positioning of the
method against the state-of-the-art is clear (Table 1) and demonstrates it is fulfilling a gap in the
field. The methodology behind the approach is relatively straightforward, and most of the novelty

comes from the implementation, including in particular 1) using conjugate gradient and GPU to
improve computational time, and 2) estimation of effect using a formal least square estimation
across thousands of variants without using a penalization. However, I have major concerns
regarding the application of the method using waist to hip ratio as an environmental exposure.
While I understand the motivation (WHR potentially captures multiple environmental effect), I
am worried using an anthropometric trait as a proxy for the environment can severely bias the
estimation of both G and GxE effect. I highly recommend using a standard environmental
exposure to demonstrate the performance of the method in real data. More details on this point
along additional comments are provided below.

1. The marginal genetic heritabilities estimated using MonsterLM in Figure 3 (BTW --tabulated
values would be useful) appear to be substantially larger than previously reported GWAS
heritability (extracted from the LDhub database) with e.g. ~ 0.30 vs 0.15 for HDL, ~ 0.19 vs 0.13
for LDL, ~ 0.25 vs 0.07 for HbA1C. The authors should derive heritability estimates and standard
deviation using existing methods (GCTA or LDscore or LDAK, etc) on the same data they use in
the MonsterLM application and discuss potential differences.

We thank the review for this suggestion. Benchmarking the heritability estimates of MonsterLM
to other heritability models using the same dataset is an important addition to the paper discussed
in Section 2 Theme 3.

Regarding MonsterLM's heritability estimates for HDL, LDL, and HbA1c, they are comparable to
mtg2, Bolt, and GRE (**Table 3**).

2. As mentioned above, treating WHR as an exposure is clearly my major concern. Adjusting for
a heritable covariate can induce a bias in the effect estimates of the regression coefficient
between the outcome and the SNP tested (a so-called collider bias, PMID= 25640676). The
impact of the bias would depend on the phenotypic correlation between WHR and the outcome
considered, and the heritabilities and co-heritabilities. It is unclear to me to which extent this
can impact the estimation of GxE and the follow-up analyses conducted by the authors
(distribution of GxE across variants, polygenic risk prediction, etc). Here are a few takes on this:

a. According to the LDhub database, several of the outcome studied have substantial and
sometimes highly significant genetic correlation with WHR (e.g. -0.525 , $P=10^{-12}$ with HDL, and
0.426 , $P= 2.2e^{-16}$ with Triglyceride), thus potentially modifying the effect of variants marginally
associated with HDL and triglyceride. Conversely, modelling a GxWHR on outcomes with small
genetic correlation with WHR can in principle induce genetic effect at variants with no marginal
effect on the primary outcome. It would be wise to check potential correlation between the
genetic effect on WHR and the marginal genetic and interaction effect estimated for each
outcome considered.

We thank the reviewer for highlighting these important considerations regarding marginal
genetic, interaction, and exposure (WHR) effects. Below we present **Supplementary Table 7**

where Pearson correlation coefficients are provided for each combination of total marginal
 genetic, interaction, and WHR effect betas. Overall, we find minimal correlation between each
 combination. Highlighted in yellow are the WHR-HDL and WHR-Triglyceride combinations the
 reviewer points out. As outlined in scenarios 9 – 12 of **Figure 3**, our updated simulations also
 address scenarios where the exposure and outcomes share a common genetic basis.

Supplementary Table 7. MonsterLM Pearson Correlation Coefficients with Marginal Genetic, Interaction, and WHR Exposure Effects

Trait	Corr(β_G, β_{WHR})	Corr(β_{GE}, β_{WHR})	Corr(β_G, β_{GE})
Apolipoprotein A	-0.00266	-0.00658	0.028173
Apolipoprotein B	-0.01722	-0.00946	0.050714
Cholesterol	-0.01693	-0.01242	0.072548
CRP	0.000326	0.001379	-0.0372
Glucose	0.007645	-0.00164	0.049059
HbA1c	0.008335	0.000121	0.062878
High Density Lipoprotein	-0.00348	-0.00514	0.038071
Low Density Lipoprotein	-0.01741	-0.01281	0.055702
Triglycerides	9.76x10 ⁻⁵	-0.00653	0.015742
Height	0.009866	0.002599	0.015919
Total Bilirubin	-0.01009	0.00417	0.006513

**Supplementary Table 7** | Each biomarker with a significant GxE estimate detected by
 MonsterLM Pearson correlation coefficient is calculated by grouping total marginal genetic
 effects and interaction effects.

b. (a) The simulation using G-E correlation (scenarios 2 & 3) is a commendable step toward
 quantifying the potential impact of using exposure with a genetic component. However, there
 is subtle, nonetheless important alternative model to consider that likely better match the use
 of WHR: when the E has no causal effect on Y, but instead simply share phenotypic variance.
 Typically $E = U + G1 + e1$ and $Y = U + G2 + e2$, where U is a non-genetic shared risk factor, G1
 and G2 are (correlated or not) genetic components, and e1 & e2 are independent residuals
 variances. A toy R simulation code is provided at the end of the review. As shown in this
 example, using an exposure not causal to Y but simply co-varying with it can induce bias on
 both G and GxE. For the GxE bias I arbitrarily used a hypothetical GxUnmeasured factor (S)
 interaction on WHR to illustrate my point. (b) A likely more realistic scenario –and my best
 guess (but more painful to check through simulation)– is that correlation between tag SNPs and
 causal ones induces similar bias. While such bias would be modest, it could be widespread,
 resulting in substantial deviation at the genome wide scale. The enrichment for GxE at SNPs
 with high LDscore might be a hint on that hypothesis.

(a) We thank the reviewer immensely for providing this sample simulation code regarding the
collider bias that can occur when modelling interactions with heritable covariates (PMID:
25640676). This prompted the inclusion of a set of scenarios where collider effects are
modelled into simulated exposures and phenotypes. Namely, where the E has no causal
effect on Y, is heritable (set at 0.20), and correlated with a collider (C) such that the (non-
zero) correlation between E and Y is entirely due to collider bias. The simulated E-Y
correlation in our collider bias scenario is greater than the highest observed correlation
between WHR and any outcome (Triglycerides; 0.33). In the collider bias simulations, the
E-Y correlation is entirely caused by the collider bias (Figure 1 top panel; page 8).
Accordingly, the simulated collider bias simulations present a rather extreme case. The
results of these simulations are discussed in Section 2 Theme 4 where we conclude that
MonsterLM is robust to strong but realistic collider scenarios modelled genome-wide.

In particular, we also included a scenario (**Figure 3**; scenario 12) where the exposure is
dependent on heterogeneous and additive genetic effects which corresponds to the scenarios
in the simulation script provided by the reviewer. The results are discussed in lines 593 –
598 of the revised manuscript:

*“Second, in the event of collider effects with a covariate that is heritable through additive*
*and heterogenous elements there could exist the possibility of some inflation for the*
*heritability component of MonsterLM (**Figure 3**; scenario 12). However, the conditions for*
*this scenario are presumed to be quite extreme and did not affect the GxE estimates.*
*Furthermore, GxE estimates have been shown to be stable when modelling collider biases*
*whereas genetic estimates were less well-controlled².”*

(b) We thank the reviewer for suggesting this potential false discovery bias. As an important
update, *“The enrichment for GxE at SNPs with high LDscore might be a hint on that*
*hypothesis”* is no longer uniformly enriched as it was in the original submission once we
revised the method and redid this analysis (**Supplementary Figure 3**; page 18). Also, as the
reviewer notes, the potential tagging for highly correlated SNPs to causal SNPs could be
subtle but real when estimated genome-wide. As such, we sought to add in a genome-wide
simulation scenario where all causal effect SNPs for GxE estimates were sampled from high
LDscore SNPs to represent an extreme case. Scenario 4 of **Figure 3** (page 6) samples all
interaction effects (β_{GE}) from the highest LD quartile of SNPs. Scenario 5 of **Figure 3** (page
6) samples all interaction effects (β_{GE}) from the highest LD quartile of SNPs and lowest
MAF quartile. Despite this, both heritability and interaction estimates remained unbiased.

c. Using the residual of the outcome adjusted for WHR (line 311-320) makes the quantification
of potential bias a little more challenging, but in principle should result in similar effect. BTW, I
might have missed it, but it wasn't clear to me whether the residual-based step was also used
in the simulation or only in the real data analyses.

We thank the reviewer for addressing the significance of the residualization step. Indeed, all
simulations use this step. The MonsterLM algorithm (**Figure 2**) is applied in its entirety the same
way for genome-wide simulation studies as in real data analyses.

We have since made an adjustment in our methods for this residualization step, but we note the
importance of employing it within the method and continue to do so. This is discussed in detail
in Section 1 and in the second reviewer’s first comment.

899 d. Altogether, I believe it would be difficult to fully exclude potential bias when using WHR as an
900 environmental exposure. And based on the available data, I cannot say whether the reported
real data results are valid, partly biased, or severely biased. Using a “standard” environmental
exposure, i.e. with a likely causal effect on the outcome would be much more convincing.

We thank the reviewer for pointing out this potential bias of using a heritable, anthropometric
environmental exposure. We agree with this suggestion to use a non-anthropometric obesogenic
standard environment exposure with limited heritability or outcome explained by exposure
variance. We add another environmental exposure, M10 (moderate exercise for 10 minutes),
with the outcomes studied to address this. We discuss this additional analysis in Section 2 Theme
4. Generally, we still observe non-null GxE estimates with a “standard” environmental exposure
without the aforementioned features that WHR possesses.

3. The authors conducted a useful simulation study to validate the performances of MonsterLM.
However, I found the scope of the simulation to be quite modest. As for any novel method, I
would expect a broader range of scenarios to be investigated, including at least two additional
analyses:

a. The simulation includes single block and 3-blocks analyses. They should also run a full
genome (all 60 blocks used in the real data analysis). I appreciate the computational burden of
such an analysis, but even a limited number of replicates (e.g. 20 instead of 100) would be
informative on the variance of the estimate

We thank the reviewer for suggesting this more robust approach to method development.
Increasing the scenario conditions and using genome-wide simulations has been a major revision
to this resubmission discussed in Section 2 Theme 2. We use 10 simulations per scenario as
generating the unique exposure and phenotype combinations combined with calculating the
heritability and GxE estimates were very computationally taxing (i.e. each point in **Figure 3** is
the computational equivalent to the real data analyses in **Figure 4a**). We find 10 simulations per
scenario provides a robust picture.

b. As the real data application suggests that contribution of GxE varies with MAF and LD (supp
Fig2), some simulations drawing effect conditional on those two parameters would help

confirming the robustness of the approach conditional on these parameters.

We thank the reviewer for this suggestion. As we noted earlier, the revised MAF and LD
 stratification analyses (**Supplementary Figure 3**; page 17) do not display any obviously
 disproportionate contribution of any strata as the original submission results did. We further
 added three more simulation scenarios that display MonsterLM robustness in different MAF/LD
 strata:

i) We refer to response 2b in the previous comment where scenario 4 and 5 of **Figure 3**
 samples GxE effect SNPs from the highest LDscore quartile and highest LDscore quartile
 plus lowest MAF quartile, respectively, illustrating robustness in an extreme condition of
 this parameter.

ii) Lines 391 – 394 of the manuscript:

*“We further stratified SNPs into 20 bins based on MAF and LD, and individually tested*
 *each stratum for genetic and interaction effects. Each stratum provided consistent*
 *estimates (after adjusting for the number of SNPs), further confirming the robustness of*
 *MonsterLM to MAF and LD (Supplementary Table 3).”*

Supplementary Table 3. The Average of 10 MonsterLM Scenario 2 Simulation Estimates Stratified by MAF and LD.

Group	SNP Count, P	LD Strata	MAF Strata	Average Estimate Per SNP R^2_G	Average Estimate Per SNP $R^2_{G \times E}$
129335	(0,0.25]	(0.05,0.10)	6.318×10^{-7}	2.481×10^{-7}
66427	(0,0.25]	(0.10,0.20]	7.580×10^{-7}	3.148×10^{-7}
28007	(0,0.25]	(0.20,0.30]	8.496×10^{-7}	3.727×10^{-7}
18256	(0,0.25]	(0.30,0.40]	8.845×10^{-7}	3.560×10^{-7}
15622	(0,0.25]	(0.40,0.50]	8.656×10^{-7}	3.537×10^{-7}
66442	(0.25,0.50]	(0.05,0.10)	9.068×10^{-7}	3.516×10^{-7}
79133	(0.25,0.50]	(0.10,0.20]	9.554×10^{-7}	3.703×10^{-7}
46717	(0.25,0.50]	(0.20,0.30]	1.133×10^{-6}	4.579×10^{-7}
34669	(0.25,0.50]	(0.30,0.40]	1.187×10^{-6}	4.875×10^{-7}
30686	(0.25,0.50]	(0.40,0.50]	1.146×10^{-6}	4.509×10^{-7}
33574	(0.50,0.75]	(0.05,0.10)	1.091×10^{-6}	4.199×10^{-7}
69734	(0.50,0.75]	(0.10,0.20]	1.061×10^{-6}	4.108×10^{-7}
57918	(0.50,0.75]	(0.20,0.30]	1.242×10^{-6}	4.853×10^{-7}
49730	(0.50,0.75]	(0.30,0.40]	1.296×10^{-6}	5.157×10^{-7}
46691	(0.50,0.75]	(0.40,0.50]	1.226×10^{-6}	4.840×10^{-7}
12840	(0.75,0.90]	(0.05,0.10)	1.187×10^{-6}	4.907×10^{-7}
48347	(0.75,0.90]	(0.10,0.20]	1.018×10^{-6}	4.047×10^{-7}
61388	(0.75,0.90]	(0.20,0.30]	1.111×10^{-6}	4.333×10^{-7}

65608	[0.75,0.90]	[0.30,0.40]	1.143×10^{-6}	4.483×10^{-5}
69464	[0.75,0.90]	[0.40,0.50]	1.057×10^{-6}	3.9292×10^{-7}

**MAF/LD Stratification Simulations.** The average estimate across 10 simulations for each MAF/LD
stratum. Each stratum estimate is divided by the number of SNPs included. Scenario 2 (See Figure
3) is applied and R^2_G , and $R^2_{G \times E}$ represent the per SNP average for heritability and GxE estimates
for each group, respectively. Non-zero effect SNPs are randomly distributed across all strata.

4. I am a puzzle with some of the R^2 estimates from figure 3. For example, for HDL, the R^2E
(~ 0.2), R^2G (~ 0.25) and $R^2G \times E$ (~ 0.6) sum up to 1+. Not sure if I am missing something,
otherwise it would mean that the variance of HDL is basically fully explained by all three terms,
which sounds unrealistic. Please clarify or revise. Full genome simulation under various
scenarios (see comment 3) would be helpful in validating the analytical standard deviation of the
approach. Note that based on the method description I do not see why there would be an issue
there, and again I am mostly worried that the large $R^2G \times E$ only reflects bias due the use of
WHR as an exposure.

We thank the reviewer for this important observation. We have attributed this finding to the
biases discussed in Section 2 Theme 1 and we outline our improvements to the method in
Section 1 and 2. Regarding the utility of analytical standard deviations, we additionally added a
section to the results about MonsterLM precision (lines 374 – 376):

*“Furthermore, observed precision estimates (i.e. variance of estimates across the 12*
*simulations) did not significantly differ from the precision estimates predicted from equations (8)*
*– (11) (Supplementary Table 2).”*

5. The reported PS are also not much in agreement with the reported $R^2G \times E$. For example, it is
challenging to explain how the GxE interaction on HDL explain >50% of the variance, but are
providing only very modest improvement in prediction as compared to the marginal genetic
effect, which itself explain “only” 25% of the variance. Again, a likely explanation is a potential
bias due to the use of WHR.

We thank the reviewer for this insightful observation. Indeed, given the amount of GxE variance
estimated in the original submission a greater improvement in polygenic score prediction would
have been expected. We have since reported a much more modest range of GxE estimates (1 -
7%) in this resubmission wherein univariate $G \times E_{WHR}$ SNPs recover a yet smaller fraction of that
(Figure 5). Accordingly, the polygenic scores now report a much more modest prediction
increase where higher GxE estimates (i.e. CRP and Triglycerides) report a greater relative
increase as compared to polygenic score only including marginal genetic effects. This is noted in
lines 517 – 528 of the manuscript:

*“Finally, we examined if the predictiveness of polygenic score of all eight outcomes with*
*significant interaction variance could be improved by incorporating interactions. To select SNPs*
*and interaction effects to be included in each PS, we used both P_G and P_{GE} thresholds of 10^{-2} ,*

10^{-3} , 10^{-4} , and 10^{-5} in the discovery set when testing either each SNP individually or both a single
SNP and corresponding interaction, respectively. Each PS was then tested in the validation
sample for association with its corresponding outcome, with twenty total P_G and P_{GE}
combinations. PS prediction R^2 was slightly improved ($p < 0.05$ for improvement) by
incorporation of interaction effects for the trait with the highest interaction variance, CRP
(**Supplementary Figure 6**), with the relative increase in prediction R^2 ranging from 0% to 1.3%
across the outcomes analyzed (for interaction significance thresholds of 10^{-4} , 10^{-5} ;
**Supplementary Table 9**). The largest increases in polygenic score predictiveness tended to occur
in traits with the largest GxE variance observed (**Figure 3**; **Supplementary Figure 6**).”

**Minor comments.**

6. Triglycerides, CRP, and HDL are presented in the abstract and used in Figure 3, but then
disappear from further figures. Some explanation or harmonization are welcome.

We thank the reviewer for helping to clarify this. The subsequent figures relate to analyses that
explore non-null GxE estimates and as such we excluded the null outcomes. This is made clearer
in the manuscript’s lines 482 – 484 now:

“Using the eight outcomes with significant GxE interaction variance (no further analyses were
conducted with outcomes having non-significant GxE interaction variance including glucose,
height, and total bilirubin), we conducted univariate linear regression ...”

7. Line 136 “As expected, all heritability estimates were significant...” may be an
overstatement?

Thank you. We modified this statement to specifically denote the heritability of blood
biomarkers that were examined, which were known to have similar heritability and shown in our
revised benchmarking estimations as noted in lines 432 – 433:

“Outcome heritability estimates for all 13 traits were significant and largely consistent with
published estimates and other methods (BOLT, mtg2, and GRE; **Tables 2 & 3**).”

8. Line 141-142: “SNPs with ... and higher LDscore to disproportionally contribute to interaction
variance”. The current consensus is that causal variants would tend to have low LDscore
(PMID= 29700474). If not explain by potential bias (see comment 4b), some discussion about
this is welcome.

Thank you for this observation and citation. We have since amended this line (449 – 450) as
follows since this analysis was updated with the revised methodology:

“The average SNP contribution to G and GxE did not markedly differ by MAF or LDscore,
confirming the absence of large differences in contribution (**Supplementary Figure 3**).”

9. A potentially interesting additional and simple metric to report to assess the link between
marginal genetic effect and interaction (Fig 4 and supp Fig3) is the correlation –i.e. $\text{cov}(\text{beta}_G,$
$\text{beta}_{G \times E})$

Thank you for this suggestion. We have included this metric by computing the Pearson
correlation coefficients for the marginal genetic effect betas to the interaction effect betas. The
results are shown in the far-right column of **Supplementary Table 7** (page 30). These are the
betas from the multiple linear regression not the univariate regressions.

10. Line 219-200: “existing methods are susceptible to bias and often yield vastly different
estimates even when applied to the same data^{10–12}.” Looking at e.g. Fig1 from Ewans et al.
(ref #12), I think that statement is a too strong. There are differences, but there are now much
better understand and the various existing corrections produce fairly similar estimates.

Thank you. Indeed, this is too strong of a statement in the original submission. We have revised
“often” to “may.”

11. I assume this is the case, but please confirm the R^2_G are derived using the residualized
outcomes (i.e. adjusted for the exposure)

Thank you. We confirm that this is indeed the case regarding additive genetic estimates and that
all genome-wide simulations follow the same algorithm as explained in the methods and **Figure**
**2**.

#####
Toy R simulation illustrating potential bias...

Thank you. We are grateful for this generous contribution of R code to illustrate potential biases.

**References:**

1. Aschard, H., Vilhjálmsson, B. J., Joshi, A. D., Price, A. L. & Kraft, P. Adjusting for
Heritable Covariates Can Bias Effect Estimates in Genome-Wide Association Studies. *Am. J.*
*Hum. Genet.* **96**, 329–339 (2015).

2. Akimova, E. T., Breen, R., Brazel, D. M. & Mills, M. C. Gene-environment
dependencies lead to collider bias in models with polygenic scores. *Sci. Rep.* **11**, 9457 (2021).

3. Dahl, A., Cai, N., Flint, J. & Zaitlen, N. GxEMM: Extending linear mixed models to
general gene-environment interactions. *bioRxiv* 397638 (2018) doi:10.1101/397638.

4. Shin, J. & Lee, S. H. GxEsum: a novel approach to estimate the phenotypic variance
explained by genome-wide GxE interaction based on GWAS summary statistics for biobank-
scale data. *Genome Biol.* **22**, 183 (2021).

5. Lee, S. H. & van der Werf, J. H. J. MTG2: an efficient algorithm for multivariate linear
mixed model analysis based on genomic information. *Bioinformatics* **32**, 1420–1422 (2016).

6. Ni, G. *et al.* Genotype–covariate correlation and interaction disentangled by a whole-
genome multivariate reaction norm model. *Nat. Commun.* **10**, 2239 (2019).

7. Speed, D. *et al.* Reevaluation of SNP heritability in complex human traits. *Nat. Genet.*
**49**, 986–992 (2017).

8. Speed, D. & Balding, D. J. SumHer better estimates the SNP heritability of complex
traits from summary statistics. *Nat. Genet.* **51**, 277–284 (2019).

9. Evans, L. M. *et al.* Comparison of methods that use whole genome data to estimate the
heritability and genetic architecture of complex traits. *Nat. Genet.* **50**, 737–745 (2018).

REVIEWERS' COMMENTS

Reviewer #1 (Remarks to the Author):

The authors have fully addressed the concerns I raised in my previous review of the manuscript. Therefore, I have no further comments to add.

Reviewer #2 (Remarks to the Author):

The authors have done a good job addressing my concerns and the other reviewers' concerns.

Reviewer #3 (Remarks to the Author):

See attached document

First of all, I would like to commend the authors for the extensive investigation on the performance and robustness of their approach. My main concern was about the very large estimated contribution of GxE to the phenotypic variance. I am glad to see much more realistic estimates from this updated version. I found the new version of the manuscript much clearer, though I still have a few comments.

1. The new Table 4 comparing the key features of the existing approaches is much useful to position the proposed method against the state-of-the-art. However, since the first submission of the manuscript, some other approaches have also been proposed. In particular, I am very curious about how MonsterLM compares against the estimated effect from LDSC regression applied to GxE chi-square. It happens that a recent paper published such an application focusing on BMI (<https://doi.org/10.1038/s42003-023-04679-4>). It should be fairly easy to run, as it just requires the GxE GWAS results. As one main argument for the MonsterLM is its ability to estimate GxE contribution from very large dataset, the LDSC approach which only used summary stat (and can therefore be applied to GWAS of very large sample size) seems a relevant additional comparison.
2. I have two comments regarding Supplementary Table 7. First β_{WHR} is defined as “Total WHR exposure effect betas”. Based on the text I assume it is the effect of WHR on the outcome tested. Would be worth make that clear. Second, the comment from the original submission was about genetic correlation between WHR and the outcome tested, which might induce bias in the GxE effect estimates. The simulation are reassuring, although, with table S7 in hands, it would be of interest to add two columns to this table to show the correlation between $\beta_{\text{WHR_SNP}}$ and both β_{g} and β_{GxE} , where $\beta_{\text{WHR_SNP}}$ is the estimated effect between the genetic variants selected and WHR. The absence of correlation would confirm the validity of the test. The presence of correlation would not necessarily invalidate the approach but might be worth mentioning for potential future investigations.
3. I assume that the GxE term described in “Step One” (line 162+) is derived as the product of the standardized G and E (so that they have both mean 0 and variance 1). Otherwise, the GxE term would be correlated to both G and E, which is undesired given the effect of GxE is estimated independently of E and G (i.e. equation (2,3,4)). If so, it is worth clarifying this step in the method. If not (i.e. G and E are not standardized), the authors should explain how this correlation is handled in the parameter estimation.
4. Figure 2 is not that easy to read and might be improved. May be by splitting the three scenarios (E and Y continuous/binary), to clarify the processing of the four variables (Y, E, G and GxE)? I am also not sure to understand some steps such as “Get i Nxm_max”. Please define each term used.
5. Regarding the difference in H2 estimates that I mentioned in my first review, I checked again and the LDSC estimates I mentioned are correct (the ldhub website is down, but the results can be found in the Table S1 from the paper, <https://doi.org/10.1093/bioinformatics/btw613>). Based to the additional comparison with other methods (BoltLMM, mtg2 and GRE, Table 3), it looks like the observed difference is not specific to MonsterLM, but might be due to the modelling choice or some data specific aspects (variability of h2 across cohort, or due to specific phenotype pre-processing). LDSC is very commonly used by the community and it might be of interest to add the results from this method in the comparison. If the differences mentioned in my first comments are confirmed, it might be worth including a bit of discussion (which can rely on the existing literature, e.g. PMID=29700474) about this.
6. Line 188 “the inversion of the U matrix”. I guess the authors mean the inversion of the (U^tU) matrix ?
7. When presenting upper and lower bound of 95% confidence interval, you may use a comma separator instead of “-” which might be confused with a minus sign!

**RE: Article third submission (ID NCOMMS-21-15930B). A versatile, fast and unbiased**
**method for estimation of gene-by-environment interaction effects on biobank-scale**
**datasets.**

**Individual Reviewers' Comments:**

**Reviewer #1 (Remarks to the Author):**

The authors have fully addressed the concerns I raised in my previous review of the
manuscript. Therefore, I have no further comments to add.

We thank the reviewer for their guidance and thorough reviews.

**Reviewer #2 (Remarks to the Author):**

The authors have done a good job addressing my concerns and the other reviewers'
concerns.

We thank the reviewer for their guidance and thorough reviews.

**Reviewer #3 (Remarks to the Author):**

First of all, I would like to commend the authors for the extensive investigation on the
performance and robustness of their approach. My main concern was about the very
large estimated contribution of GxE to the phenotypic variance. I am glad to see much
more realistic estimates from this updated version. I found the new version of the
manuscript much clearer, though I still have a few comments.

We thank the reviewer for this commendation and further suggestions.

1. (a) The new Table 4 comparing the key features of the existing approaches is much
useful to position the proposed method against the state-of-the-art. However, since the
first submission of the manuscript, some other approaches have also been proposed. In
particular, I am very curious about how MonsterLM compares against the estimated
effect from LDSC regression applied to GxE chi-square. It happens that a recent paper
published such an application focusing on BMI ([https://doi.org/10.1038/s42003-023-](https://doi.org/10.1038/s42003-023-04679-4)
[04679-4](https://doi.org/10.1038/s42003-023-04679-4)). (b) It should be fairly easy to run, as it just requires the GxE GWAS results. As
one main argument for the MonsterLM is its ability to estimate GxE contribution from
very large dataset, the LDSC approach which only used summary stat (and can therefore
be applied to GWAS of very large sample size) seems a relevant additional comparison.

 We thank the reviewer for their comments and suggestions. We have labelled this comment with
 sub headers (a) - (b).

 (a) The reviewer raises a fitting point about adding another novel GxE application (based on
 the LDSC regression heritability method) to our **Table 4** which compares MonsterLM
 GxE to the state-of-the-art. The aforementioned paper, Jung et al., 2023¹, appears to
 directly adapt LDSC regression to estimate GxE by using GxE GWAS in place of
 additive SNP effects. While the study is limited in its assessment of model performance
 (as a direct GxE LDSC adaptation is a novel implementation) and robustness we can infer
 the advantages and limitations compared to MonsterLM based on the well-studied
 features of LDSC regression. This comparison has been added to **Table 4** in the main text
 and is shown in a subset below:

Table 4 subset. Comparison of current methods estimating gene-by-environment contributions to MonsterLM

Method	Description	Advantages	Limitations	MonsterLM
LDSC GxE ¹	Estimates genome-wide GxE variance using GxE GWAS (GWIS) summary statistics. GWIS replaces GWAS in standard LDSC regression ² to estimate h_{GxE}^2 .	Utilizes GWAS summary statistics; computationally efficient; typically, robust to confounding from stratification and common environmental effects.	Risk of h^2 underestimation in high LD or low polygenicity regions; bias when LD scores from reference population and GWAS mismatch.	Individual-level data advantages include better handling of LD and complex interactions; robust in low polygenicity scenarios.

 (b) We have applied the LDSC GxE method through generating GxE betas with WHR as an
 exposure for the outcome-WHR combinations shown in **Table 3**. A subset of **Table 3**
 heritability and GxE model comparisons for MonsterLM and LDSC (in red text) is shown
 below:

Trait	MonsterLM h_G^2 (σ)	LDSC h_G^2 (σ)	MonsterLM $R^2_{GxE_{WHR}}$ (σ)	LDSC $R^2_{GxE_{WHR}}$ (σ)
Apolipoprotein A	0.281 (0.0046)	0.217 (0.0320)	0.024 (0.0043)	0.003 (0.0021)
Apolipoprotein B	0.219 (0.0045)	0.129 (0.0246)	0.024 (0.0043)	0.010 (0.0020)
Cholesterol	0.179 (0.0045)	0.122 (0.0181)	0.018 (0.0042)	0.014 (0.0023)
CRP	0.238 (0.0046)	0.056 (0.0079)	0.071 (0.0044)	0.003 (0.0020)
Glucose	0.105 (0.0047)	0.045 (0.0044)	0.005 (0.0046)	0.009 (0.0041)
HbA1c	0.289 (0.0047)	0.113 (0.0072)	0.009 (0.0044)	0.018 (0.0041)

61	HDL-Cholesterol	0.313	0.246	0.038	0.007
62		(0.0044)	(0.0329)	(0.0041)	(0.0021)
63	LDL-Cholesterol	0.173	0.115	0.019	0.014
64		(0.0045)	(0.0226)	(0.0043)	(0.0023)
65	Triglycerides	0.205	0.163	0.068	0.018
66		(0.0042)	(0.0189)	(0.0041)	(0.0041)
67	Height	0.683	0.447	0.001	0.001
68		(0.0045)	(0.0202)	(0.0039)	(0.0020)
69	Total Bilirubin	0.399	0.087	0.001	0.000
70		(0.0046)	(0.0348)	(0.0043)	(0.0017)

Generally, LDSC heritability and GxE estimates were lower than MonsterLM. LDSC estimates
are now reported in the results (lines 488 – 490):

*“The LDSC GxE analysis used the full participant list, SNP set, and phenotypic data as in
MonsterLM. GxE estimates were lower than MonsterLM for all outcomes.”*

A discussion of the LDSC GxE estimates compared to MonsterLM occurs in lines 572 – 584 in
the main text:

*“When comparing MonsterLM GxE estimates to LDSC GxE, MonsterLM estimated 8 of 11
outcomes to be non-null and LDSC GxE estimated 7 of 11 outcomes to be non-null. LDSC GxE
estimates ranged from 0 – 1.8% and were lower than MonsterLM for each outcome (as was
consistent with LDSC heritability estimates versus MonsterLM, Bolt, and GRE). Some LDSC
GxE advantages include the fast computational speed of summary level statistics compared to
individual-level data and robustness to stratification and common environmental effects.
However, the potential for LDSC underestimation is a discussed limitation in the literature. For
example, Evans et al., 2018³ conducted a heritability model comparison study where they
showed a limitation of LD score regression was its potential to underestimate h^2 if the trait is
not highly polygenic (such as in the case of total bilirubin; **Table 3**). Furthermore, consistently
smaller LDSC h^2 estimates have been shown when compared to GREML in the same data set⁴
and as the lowest estimate in a recent protocols study compared to 10 other approaches
(including GREML, LDAK, threshold GRMs, and SumHer)⁵.”*

2. (a) I have two comments regarding Supplementary Table 7. First β_{WHR} is defined
as “Total WHR exposure effect betas”. Based on the text I assume it is the effect of
WHR on the outcome tested. Would be worth make that clear. (b) Second, the
comment from the original submission was about genetic correlation between WHR
and the outcome tested, which might induce bias in the GxE effect estimates. The
simulation are re- assuring, although, with table S7 in hands, it would be of interest to
add two columns to this table to show the correlation between $\beta_{\text{WHR_SNP}}$ and
both β_{g} and β_{GxE} , where $\beta_{\text{WHR_SNP}}$ is the estimated effect between the
genetic variants selected and WHR. The absence of correlation would confirm the
validity of the test. The presence of correlation would not necessarily invalidate the

approach but might be worth mentioning for potential future investigations.

We thank the reviewer for requesting clarification on this topic. We have labelled this comment
with sub headers (a) - (b).

(a) “Total WHR exposure effect betas” was referring to the $1 \times m$ row vector of regression
coefficients from the MonsterLM heritability model of WHR (where WHR is the
outcome). It is not the regression coefficients from WHR as regressor to an outcome
regressand. We thank the reviewer for pointing this out as the previous manuscript
description “WHR exposure effects” is misleading and has been revised to “WHR
heritability effects.” A revised description of the correlation results (main text lines 463 –
469) and **Supplementary Table 7** (revisions in red) is shown below:

“Consistent with the directionality concordance for each outcome at $P_G < 1$ and $P_{GE} < 1$,
Pearson correlation coefficients of estimated genetic regression coefficients for each outcome,
$\hat{\beta}_{1 \times m}$ (m is the number of SNPs: 1,030,579), were significant for all outcomes in **Figure 4b** for
$\hat{\beta}_G$ and $\hat{\beta}_{GE}$ (**Supplementary Table 7**). When extending the Pearson correlation tests to estimated
genetic regression coefficients from WHR heritability (WHR heritability; **Supplementary Table**
**6**), $\hat{\beta}_{h^2_{WHR}}$, neither $\hat{\beta}_G$ or $\hat{\beta}_{GE}$ were significantly correlated with $\hat{\beta}_{h^2_{WHR}}$ for almost all outcomes
(**Supplementary Table 7**).”

Supplementary Table 7. MonsterLM Pearson Correlation Coefficients with Genetic, Interaction, and WHR Heritability Effects

Trait	Corr($\hat{\beta}_G, \hat{\beta}_{h^2_{WHR}}$)	Corr($\hat{\beta}_{GE}, \hat{\beta}_{h^2_{WHR}}$)	Corr($\hat{\beta}_G, \hat{\beta}_{GE}$)
Apolipoprotein A	-0.00266	-0.00658	0.028173*
Apolipoprotein B	-0.01722*	-0.00946	0.050714*
Cholesterol	-0.01693*	-0.01242	0.072548*
CRP	0.000326	0.001379	-0.0372*
Glucose	0.007645	-0.00164	0.049059*
HbA1c	0.008335	0.000121	0.062878*
HDL-Cholesterol	-0.00348	-0.00514	0.038071*
LDL-Cholesterol	-0.01741*	-0.01281*	0.055702*
Triglycerides	9.76×10^{-5}	-0.00653	0.015742*
Height	0.009866	0.002599	0.015919*
Total Bilirubin	-0.01009	0.00417	0.006513

**Pearson Correlation Coefficients with Genetic, Interaction, and WHR (exposure) heritability regression**
**coefficients.** For each outcome calculated in MonsterLM, a Pearson correlation test is performed
between each pair of regression coefficients ($\hat{\beta}_{1 \times m}$) in the three combinations above. $\hat{\beta}_G, \hat{\beta}_{GE}, \hat{\beta}_{h^2_{WHR}}$:
genetic regression coefficients, interaction regression coefficients, and WHR heritability regression
coefficients, respectively. Significant Pearson correlation tests at $*p < 0.05/11$.

(b) Part (a) now clarifies that the previous manuscript version’s “ β_{WHR} ” is a $1 \times m$ row

vector of WHR heritability regression coefficients now denoted as “ $\hat{\beta}_{h^2_{WHR}}$.” The first two
columns in the above table compare the Pearson correlation between $\hat{\beta}_{h^2_{WHR}}$ and $\hat{\beta}_{GE}$ and
$\hat{\beta}_{h^2_{WHR}}$ and $\hat{\beta}_G$. The results show that most of the correlations are not significant and for
outcomes with the largest GxE estimates, CRP and Triglycerides, the Pearson correlation
coefficient is not significant in all the combinations tested with $\hat{\beta}_{h^2_{WHR}}$. These results
generally support the reviewer’s point that “*The absence of correlation would confirm*
*the validity of the test.*”

3. I assume that the GxE term described in “Step One” (line 162+) is derived as the
product of the standardized G and E (so that they have both mean 0 and variance 1).
Otherwise, the GxE term would be correlated to both G and E, which is undesired given
the effect of GxE is estimated independently of E and G (i.e. equation (2,3,4)). If so, it is
worth clarifying this step in the method. If not (i.e. G and E are not standardized), the
authors should explain how this correlation is handled in the parameter estimation.

We thank the reviewer for suggesting further clarification on the order of operations regarding
“Step One” Genotype and Phenotype Input and Quality Control. Indeed, “G” and “E” are
normalized so that mean = 0 and variance = 1. Line 171 – 177 in the main text has been edited
(in red) so that it is clearer that the G and E matrices have been normalized *before* they are
multiplied to create the GxE product:

“*The standard linear model for an outcome, Y, when an interaction term is included can be*
*expressed as:*

$$Y = \beta_G G + \beta_E E + \beta_{GE} GE + \epsilon \quad (1)$$

*Where G is the standardized genotype matrix, E is the quantile normalized environmental*
*exposure, GE is the product between each genotype matrix and environmental exposure,*
*resulting in a matrix with the same dimensionality as G. G is coded in the additive model*
*({0,1,2}) and standardized so that the mean = 0 and standard deviation = 1 for each SNP. GE is*
*the quantile normalized product of G and E.”*

Furthermore, lines 177 – 184 describe the specific set of transformations applied to E, Y, and G
before E and G are multiplied to generate the GxE term. In **Figure 2**, “Step 1” has been revised
so that the exact sequence of transformations for each parameter is apparent. See revised **Figure**
**2** in comment #4 below.

4. Figure 2 is not that easy to read and might be improved. May be by splitting the three
scenarios (E and Y continuous/binary), to clarify the processing of the four variables
(Y, E, G and GxE)? I am also not sure to understand some steps such as “Get i
Nxm_max”. Please define each term used.

We thank the reviewer for suggesting further improvement of the step-by-step methodology
figure for MonsterLM. We have opted to retain the **Figure 2** format with the three continuous

and dichotomous scenario-specific transformations to keep this pipeline figure in the main text
(not supplementary) in accordance with Nature Communications author guidelines of no more
than 10 main text figures/tables. However, we have made two adjustments within the figure to
improve the conceptualization of the order of “Step 1” transformations and for clearer definitions
of each term:

- i) We have extended the within-figure legend (footnote) to define each term not
explained within the figure text (and simplified the G matrices description).
Furthermore, the **Figure 2** legend in the main text now defines each term.
 - ii) We have made the order of transformations for “Step 1” clearer by labelling
numerical indicators that suggest the sequence of functions for each parameter. We
have also added an additional disclaimer at the figure footnote to denote which
functions are unique to the parameter scenario.

The revised **Figure 2** with its caption is shown below:

Figure 2

**Figure 2** | The MonsterLM method split into three steps for continuous outcomes and exposures.
The first step describes data processing, the second step describes methods of computing least
squares, and the third step describes how to finalize estimates and compute confidence intervals.
Sections outlined by blue, red, or green are transformations only to be applied with that variable-
type combination described in the footnote. *E*: exposure matrix; *Y*: outcome matrix; *G*: genotype
matrix; *GxG*: interaction matrix. *N*; number of participants; *M*; number of SNPs; m_{\max} : maximum
number of SNPs to be partitioned genotype matrix.

5. Regarding the difference in H2 estimates that I mentioned in my first review, I
checked again and the LDSC estimates I mentioned are correct (the ldhub website is
down, but the results can be found in the Table S1 from the paper,
<https://doi.org/10.1093/bioinformatics/btw613>). Based to the additional comparison
with other methods (BoltLMM, mtg2 and GRE, Table 3), it looks like the observed
difference is not specific to MonsterLM, but might be due to the modelling choice or
some data specific aspects (variability of h2 across cohort, or due to specific
phenotype pre-processing). LDSC is very commonly used by the community and it
might be of interest to add the results from this method in the comparison. If the
differences mentioned in my first comments are confirmed, it might be worth including
a bit of discussion (which can rely on the existing literature, e.g. PMID=29700474)
about this.

We thank the reviewer for this suggestion. Indeed, LDSC regression is commonly used in
the community and offers certain advantages as discussed in comment #1. Therefore, a
worthwhile comparison with MonsterLM heritability is to run LDSC regression on the
outcomes in our study using the LDSC method as advertised (with the same dataset used in
the main text). The results are added to **Table 3** and a subset of that table comparing
MonsterLM to LDSC can be referred to in comment #1. As the differences in estimation
the reviewer mentions are noted, discussion has been added in comment #1.

6. Line 188 “the inversion of the U matrix”. I guess the authors mean the inversion of the
($U^T U$) matrix ?

Thank you. This has been revised in lines 196 – 197:

*“The calculation is done such that the only practical limitation is the speed of the inversion of*
*the $U^T U$ matrix, without any restriction on n .”*

7. When presenting upper and lower bound of 95% confidence interval, you may
use a comma separator instead of “-” which might be confused with a minus sign!

Thank you. All instances of the “-” separator in main text and supplemental tables
have been replaced with a comma.

References:

1. Jung, H.-U. *et al.* Gene-environment interaction explains a part of missing heritability in human body mass index. *Commun. Biol.* **6**, 1–11 (2023).
2. Bulik-Sullivan, B. K. *et al.* LD Score regression distinguishes confounding from polygenicity in genome-wide association studies. *Nat. Genet.* **47**, 291–295 (2015).
3. Evans, L. M. *et al.* Comparison of methods that use whole genome data to estimate the heritability and genetic architecture of complex traits. *Nat. Genet.* **50**, 737–745 (2018).
4. Ni, G., Moser, G., Schizophrenia Working Group of the Psychiatric Genomics Consortium, Wray, N. R. & Lee, S. H. Estimation of Genetic Correlation via Linkage Disequilibrium Score Regression and Genomic Restricted Maximum Likelihood. *Am. J. Hum. Genet.* **102**, 1185–1194 (2018).
5. Srivastava, A. K., Williams, S. M. & Zhang, G. Heritability Estimation Approaches Utilizing Genome-Wide Data. *Curr. Protoc.* **3**, e734 (2023).